# MDN brain descending neurons coordinately activate backward and inhibit forward locomotion

**Arnaldo Carreira-Rosario[1†], Aref Arzan Zarin[1†], Matthew Q Clark[1†‡], Laurina Manning[1], Richard D Fetter[2], Albert Cardona[2], Chris Q Doe[1*]**

[1]Institute of Neuroscience, Howard Hughes Medical Institute, University of Oregon, Eugene, United States; [2]Janelia Research Campus, Howard Hughes Medical Institute, Ashburn, United States

**Abstract** Command-like descending neurons can induce many behaviors, such as backward locomotion, escape, feeding, courtship, egg-laying, or grooming (we define 'command-like neuron' as a neuron whose activation elicits or 'commands' a specific behavior). In most animals, it remains unknown how neural circuits switch between antagonistic behaviors: via top-down activation/ inhibition of antagonistic circuits or via reciprocal inhibition between antagonistic circuits. Here, we use genetic screens, intersectional genetics, circuit reconstruction by electron microscopy, and functional optogenetics to identify a bilateral pair of *Drosophila* larval 'mooncrawler descending neurons' (MDNs) with command-like ability to coordinately induce backward locomotion and block forward locomotion; the former by stimulating a backward-active premotor neuron, and the latter by disynaptic inhibition of a forward-specific premotor neuron. In contrast, direct monosynaptic reciprocal inhibition between forward and backward circuits was not observed. Thus, MDNs coordinate a transition between antagonistic larval locomotor behaviors. Interestingly, larval MDNs persist into adulthood, where they can trigger backward walking. Thus, MDNs induce backward locomotion in both limbless and limbed animals.
DOI: https://doi.org/10.7554/eLife.38554.001

**\*For correspondence:**
cdoe@uoneuro.uoregon.edu

[†]These authors contributed equally to this work

**Present address:** [‡]Division of Biology and Biological Engineering, California Institute of Technology, Pasadena, United States

**Competing interests:** The authors declare that no competing interests exist.

## Introduction

Animals typically execute one behavior to the exclusion of all other possible behaviors (*Briggman and Kristan, 2008*). For example, leeches can either crawl or swim, but cannot do both simultaneously (*Briggman and Kristan, 2006*; *Kristan, 2008*); or using the same set of muscles, a locust is capable of either walking or flying but cannot execute both behaviors at the same time (*Ramirez and Pearson, 1988*). Such mutually exclusive choice of behavior has also been observed in several other systems, including *Caenorhabditis elegans* (forward vs backward crawling), *Tritonia* (crawling vs swimming), leech (feeding vs swimming), tadpole (struggling vs swimming), turtle (swimming vs scratching), and zebrafish (left vs right escape) (*Berkowitz, 2002*; *Gaudry and Kristan, 2010*; *Koyama et al., 2016*; *Popescu and Frost, 2002*; *Roberts et al., 2016*; *Soffe, 1993*). The selection of a locomotor program to the exclusion of all others is necessary to prevent injury and escape predation. Despite the paramount importance of rapid transitions between antagonistic motor programs, the underlying circuitry is only beginning to be understood in *C. elegans* (*Lindsay et al., 2011*; *Piggott et al., 2011*; *Roberts et al., 2016*).

Command-like neurons can elicit specific behaviors, such as forward locomotion, backward locomotion, pausing, escape, flight, grooming, feeding, courtship, egg-laying or sleep (*Bidaye et al., 2014*; *Bouvier et al., 2015*; *Hägglund et al., 2010*; *Hampel et al., 2015*; *Hedwig, 2000*; *Hückesfeld et al., 2015*; *Kallman et al., 2015*; *Liu and Fetcho, 1999*; *Ohyama et al., 2015*;

**eLife digest** When we choose to make one kind of movement, it often prevents us making another. We cannot move forward and backward at the same time, for example, and a horse cannot simultaneously gallop and walk. These 'antagonistic' behaviors often use the same group of muscles, but the muscles contract in a different order. This requires exquisite control over muscle contractions.

Neurons located in the central nervous system form circuits to produce distinct patterns of muscle contractions and to switch between these patterns. Smooth, rapid switching between behaviors is important for animal escape and survival, as well as for performing fine movements. However, we know little about how the activity of the neuronal circuits enables this.

Carreira-Rosario, Zarin, Clark et al. set out to identify the underlying neuronal circuitry that allows larval fruit flies to transition between crawling forward and backward. Results from a combination of genetics and microscopy techniques revealed that a neuron called the Mooncrawler Descending Neuron (MDN) induces a switch from forward to backward travel. MDN activates a neuron that stops the larvae crawling forward, and at the same time activates a different neuron that is only active when the larvae crawl backward. Carreira-Rosario et al. also found that MDN triggers backward crawling in the six-limbed adult fly.

Understanding how a single neuron – in this case MDN – can trigger a smooth switch between opposing behaviors could be beneficial for the medical and robotics fields. In the medical field, understanding how movement is generated could help to improve therapies that fix damage to the relevant neuronal circuits. Understanding how behavioral transitions occur may also help to design autonomous robots that can navigate complex terrain.

DOI: https://doi.org/10.7554/eLife.38554.002

*Pearson et al., 1985*; *Sen et al., 2017*; *Tanouye and Wyman, 1980*; *von Philipsborn et al., 2011*; *Weber et al., 2015*; *Wu et al., 2015*). However, much less is known about how antagonistic motor programs are suppressed during command neuron-induced behavior. On one hand, there could be a high degree of reciprocal inhibition between neurons in antagonistic circuits; on the other hand, the command neurons that activate one behavior may also suppress antagonistic behaviors (in which case there could be minimal reciprocal inhibition). Here, we use the *Drosophila* larva to characterize the neural circuits coordinately regulating two antagonistic behaviors: forward versus backward locomotion.

*Drosophila* larva have many distinct behaviors (*Vogelstein et al., 2014*), but forward locomotion is the default locomotor behavior (*Berni et al., 2012*) and consists of coordinated posterior-to-anterior waves of somatic body wall muscle contractions driven by corresponding waves of motor neuron activity within the segmented ventral nerve cord (VNC) (*Clark et al., 2018*; *Heckscher et al., 2012*; *Hughes and Thomas, 2007*; *Pulver et al., 2015*). There are ~35 motor neurons per bilateral hemisegment, innervating 30 body wall muscles (*Landgraf and Thor, 2006*), about 250 interneurons per hemisegment (*Rickert et al., 2011*), and an unknown number of ascending and descending neurons traversing each segment of the VNC. The circuits for motor wave propagation (*Fushiki et al., 2016*), the coordination of muscle groups within each segment (*Zwart et al., 2016*), and the bilateral adjustment of muscle contraction amplitude (*Heckscher et al., 2015*) have been recently investigated; however, much less is known about the circuits promoting backward locomotion, or the switching from forward to backward locomotion.

Larvae initiate backward locomotion upon encountering a barrier or experiencing mild noxious stimulation to the anterior body (*Kernan et al., 1994*; *Robertson et al., 2013*; *Takagi et al., 2017*; *Titlow et al., 2014*; *Tracey et al., 2003*). Backward locomotion consists of anterior-to-posterior waves of motor neuron and muscle activity (*Heckscher et al., 2012*; *Pulver et al., 2015*). A segmentally reiterated VNC neuron that triggers backward locomotion has been identified (*Takagi et al., 2017*), but high-order command-like neurons for backward locomotion and the circuit for executing backward wave propagation while simultaneously suppressing forward waves remain unknown.

Here, we identify a bilateral pair of *Drosophila* brain descending neurons that coordinately activate backward locomotion and suppress forward locomotion, and identify the downstream premotor circuitry effecting the switch. Surprisingly, immortalization of CsChrimson (Chrimson) expression in these larval command-like neurons reveals that they survive metamorphosis, have the exact morphology of previously described adult 'moonwalker' neurons (*Bidaye et al., 2014*), and can induce backward walking in the adult. By analogy to the adult naming scheme, we refer to these larval brain neurons as 'mooncrawler descending neurons' (MDNs). We reconstruct the larval MDNs in an electron microscopy volume comprising the whole central nervous system (*Ohyama et al., 2015*), in which we also map its postsynaptic neuron partners. We identify the circuit motifs by which MDNs induce backward locomotion while simultaneously suppressing forward locomotion. The MDNs project their axons along the length of the nerve cord, where they directly activate an excitatory cholinergic pre-motor neuron (A18b) that is specifically active during backward waves. In parallel, the MDNs synapse onto a GABAergic inhibitory neuron (Pair1) that directly inhibits cholinergic pre-motor neurons (A27h) active specifically during forward locomotion (*Fushiki et al., 2016*); optogenetic experiments showed that MDNs activate Pair1 neurons, which then inhibit A27h and block forward locomotion. The circuit structure therefore suggests that two behaviors such as forward and backward peristaltic locomotion can maintain mutually exclusive activity due to top-down excitation/inhibition, rather than reciprocal inhibition. We conclude that the MDNs promote backward locomotion at all stages of the *Drosophila* life cycle: from the limbless crawling maggot to the limbed walking adult.

## Results

### Identification of brain neurons sufficient and necessary for larval backward locomotion

We previously showed that activating neurons labeled by the Janelia R53F07-Gal4 line could induce backward larval locomotion, but this line has broad expression in the brain, subesophageal zone (SEZ), and both motor neurons and interneurons of the VNC (*Clark et al., 2016*). To identify the neurons within this population that can induce backward locomotion, we used intersectional genetics (*Dolan et al., 2017*; *Luan et al., 2006*) to find lines labeling small subsets of the original population. We identified three lines called Split1, Split2, and Split3 labeling different subsets of the original pattern; the only neurons present in all three Split lines are a bilateral pair of neurons with cell bodies located in the ventral, anterior, medial brain with descending processes to A3-A5 in the VNC (*Figure 1A–C*, arrowheads).

All three Split lines could induce backward locomotion following Chrimson expression and activation (*Figure 1D*, *Videos 1* and *2*). Neuronal activation immediately switched locomotion from forward to backward (*Figure 1E,F*), without a significant change in the number of peristaltic waves per second (Split1, 0.48; Split2, 0.50; Split3, 0.65 before activation; Split1, 0.48; Split2, 0.56; Split3, 0.56 after activation). Conversely, using Split2 or Split3 to express the light-inducible neuronal silencer GtACR1 (*Mohammad et al., 2017*) significantly reduced backward locomotion induced by a noxious head poke (*Figure 1G,H*). It is likely that these activation and silencing phenotypes arise from the pair of ventral, anterior, medial brain descending neurons common to all three lines, although it is possible that there are different neurons in each Split line that can induce backward locomotion. We distinguish between these alternatives in the next section.

### A single pair of brain neurons can induce a switch from forward to backward locomotion

To determine whether Chrimson expression in just one or two of the ventral, anterior, medial brain neurons is sufficient to induce backward locomotion, we stochastically expressed Chrimson:Venus within the Split2 pattern via the 'FLP-out' method (*Figure 2A*). We screened populations of larvae for Chrimson-induced backward locomotion (obtaining 1–2 larvae per 100 screened), and stained the CNS to identify the Chrimson:Venus$^+$ neurons that were sufficient to induce backward locomotion. All larvae with a backward locomotion phenotype (n = 10) expressed Chrimson:Venus in one or both neurons from the anterior, medial pair that had descending projections to A3-A5 (three examples shown in *Figure 2B–D*). Conversely, all larvae that lacked Chrimson-induced backward

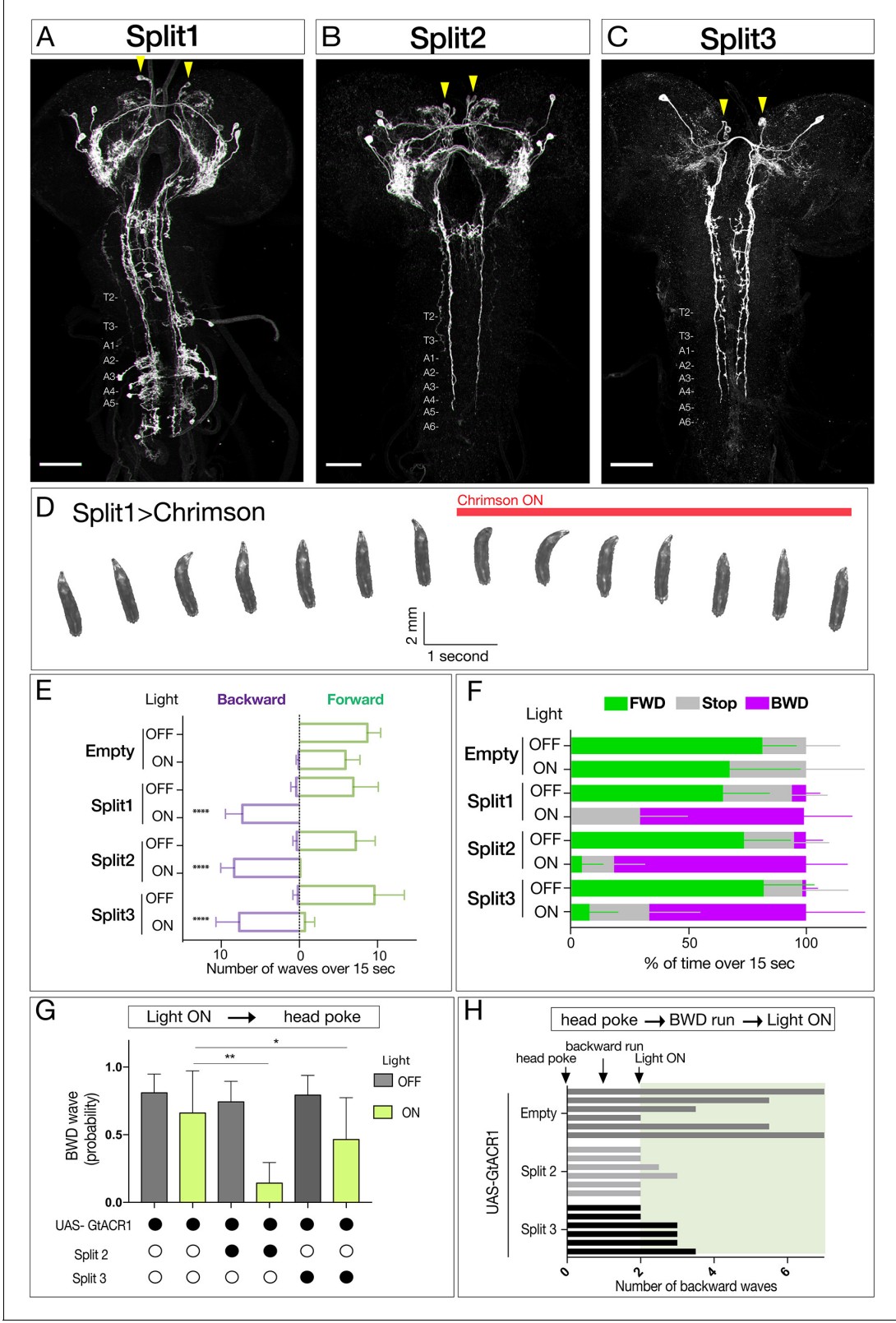

**Figure 1.** Neurons sufficient to induce backward larval locomotion. (**A–C**) Split1-Split3 lines driving expression of membrane localized Venus in the third instar CNS. Corazonin (not shown) labels a single neuron in segments T2-A6, and was used to identify VNC segment identity. The only neurons potentially common to all three lines are a pair of bilateral ventral, anterior, medial neurons (arrowheads). Maximum intensity projection of entire CNS shown. Anterior, up; scale bars, 50 µm. Genotypes: *R49F02-Gal4^{AD} R53F07-Gal4^{DBD} UAS-Chrimson:mVenus* (Split1); *R49F02-Gal4^{AD} R53F07-Gal4^{DBD}*

*Figure 1 continued on next page*

*Figure 1 continued*

*tsh-lexA lexAop-killer zipper UAS-Chrimson:mVenus* (Split2); *ss01613-Gal4 UAS-Chrimson:mVenus* (Split3). (D) Split1 activation induces backward locomotion. Genotype: *R49F02-Gal4$^{AD}$ R53F07-Gal4$^{DBD}$ UAS-Chrimson:mVenus.* (E) Split1, Split2, or Split3 activation induces backward locomotion. Number of backward or forward waves in third instar larvae over 15 s with or without Chrimson activation. N = 10 for all genotypes. Genotypes: *pBD-Gal4 UAS-Chrimson:mVenus* (Empty) and see A-C above for Split1-3 genotypes. (F) Split1, Split2, or Split3 activation induces backward locomotion. Percentage of time performing forward locomotion (green), backward locomotion (magenta) or paused (grey) in third instar larvae over 15 s with or without Chrimson activation. N = 5 for all genotypes. Genotypes, see A-C. (G) Split2 or Split3 silencing reduces initiation of backward locomotion. Backward waves induced by a noxious head poke, with or without active GtACR1. Genotypes: *pBD-Gal4 UAS-GtACR1:mVenus* (first two bars, n = 20), *R49F02-Gal4$^{AD}$ R53F07-Gal4$^{DBD}$ tsh-lexA lexAop-killer zipper UAS-GtACR1:mVenus* (middle two bars, n = 8), *ss01613-Gal4 UAS-GtACR1:mVenus* (last two bars, n = 25). (H) Split2 or Split3 neuron silencing stops ongoing backward locomotion. After each larva initiated a backward run (two backward waves), light was used to activate GtACR1 or a no Gal4 control, and the number of backward waves was counted. n = 6 for both groups; each bar represents the average of two trials for the same larva. See G for genotypes.

DOI: https://doi.org/10.7554/eLife.38554.003

locomotion (n = 20) never showed Chrimson:Venus expression in the ventral, anterior, medial descending neurons (data not shown). Based on similarity to the 'moon<u>walker</u>' neuron adult backward walking phenotype (*Bidaye et al., 2014*), we name this bilateral pair of neurons the 'moon-<u>crawler</u>' descending neurons (MDNa and MDNb), subsequently called MDNs. The MDNs are likely to be excitatory, as they are cholinergic (*Figure 2E*). We conclude that activation of as few as two of the four MDNs (either both in the same brain lobe or one in each brain lobe) is sufficient to induce a behavioral switch from forward to backward locomotion.

To determine if a short pulse of MDN activation can trigger one or more backward waves, we provided a brief 300 ms Chrimson activation of MDN and assayed the number of backward waves induced in both intact larvae and fictive CNS. We found that both the intact larvae and fictive CNS invariably performed a backward motor wave in response to a pulse of MDN activation, with the fictive CNS occasionally generating a second wave at variable times during the 25 s after Chrimson activation (*Figure 2F*). We next asked, if forced activation of MDNs can induce backward locomotion, perhaps the MDNs are normally active specifically during backward locomotion. To test this hypothesis, we used CaMPARI to monitor MDN activity during forward versus backward locomotion within the intact crawling larva. CaMPARI undergoes an irreversible green-to-red conversion upon coincident exposure to elevated Calcium (i.e. neuronal activity) and 405 nm illumination (*Fosque et al., 2015*). We used Split2 to express CaMPARI in MDNs and exposed crawling larvae to

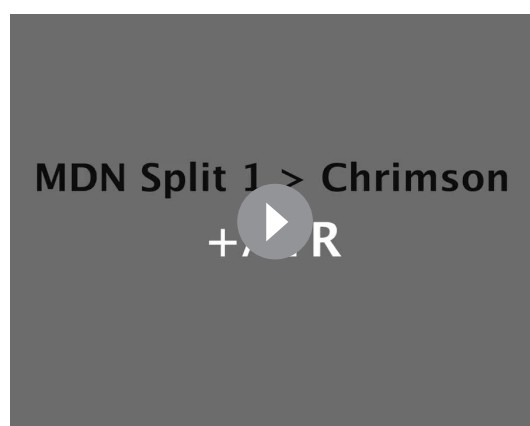

**Video 1.** MDN activation induces backward larval locomotion. Crawling behavior of third instar larvae expressing Chrimson in MDNs (Split1 > Chrimson: mVenus) with ATR. During the first 15 s, the animals are not under optogenetic light followed by 15 s under 0.5 mW/mm² of green light.
DOI: https://doi.org/10.7554/eLife.38554.004

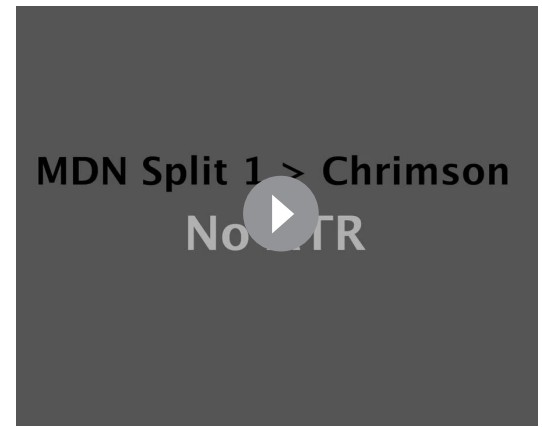

**Video 2.** MDN activation induces backward larval locomotion. Crawling behavior of third instar larvae expressing Chrimson in MDNs (Split1 >Chrimson: mVenus) without ATR. During the first 15 s, the animals are not under optogenetic light followed by 15 s under 0.5 mW/mm² of green light.
DOI: https://doi.org/10.7554/eLife.38554.005

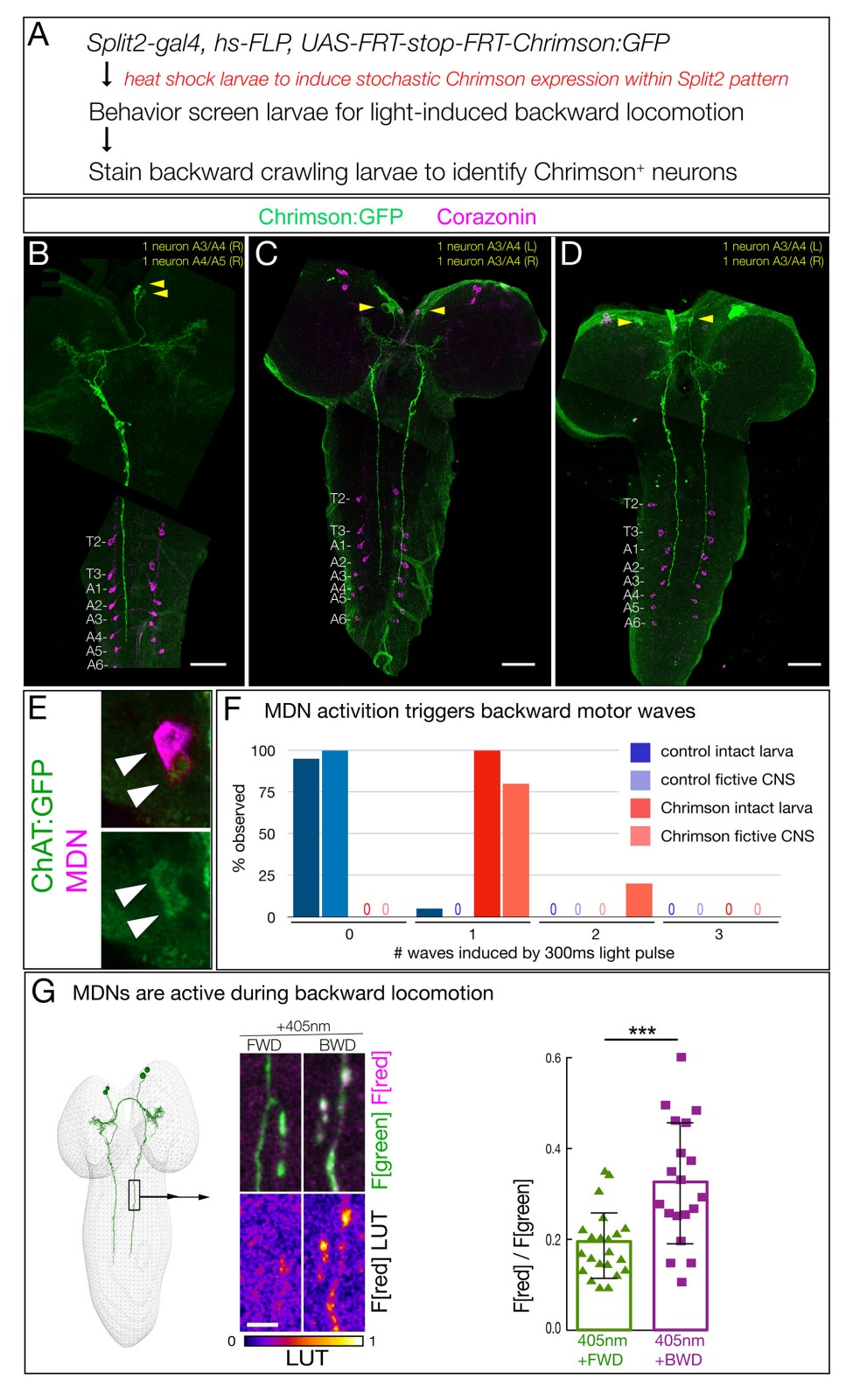

**Figure 2.** Two brain descending neurons are sufficient to induce backward larval locomotion. (**A**) Experimental flow for generating sparse, stochastic patterns of Chrimson in subsets of the Split2 expression pattern. Genotype: *hsFlpG5.PEST R49F02-Gal4[AD] R53F07-Gal4[DBD] tsh-lexA, lexAop-killer zipper UAS.dsFRT.Chrimson:mVenus*. (**B–D**) The CNS from three larvae that crawled backward in response to Chrimson activation. All show expression in neurons with medial cell bodies, bilateral arbors, and a contralateral descending projection to A3-A5. (**B**) Note there is a tear in the CNS near

*Figure 2 continued on next page*

*Figure 2 continued*

segment T1. Chrimson:Venus, green; Corazonin (Crz; segmental marker), magenta. Scale bar, 50 µm. (**E**) MDNs are cholinergic. MDNs marked with mCherry (magenta) express ChAT:GFP (green). Genotype: *R49F02-Gal4^AD^,UAS-Chrimson:mCherry; R53F07-Gal4^DBD^, mimic ChAT:GFP*. (**F**) A brief pulse of MDN activity can trigger a backward wave. Intact larvae: individual L3 larvae were subjected to 300 ms of 561 nm light and the number of backward waves was quantified (n = 20). Genotype: *ss01613-Gal4 (Split3); UAS-Chrimson::mVenus*. Fictive CNS: isolated L3 CNS was subjected to 300 ms 561 nm light and the response of the downstream neuron A18b was monitored for a backward wave of GCaMP6f activity. All MDN activations led to at least one backward wave, but there are also backward waves that occur independently of MDN activation (see Discussion), and these may account for the observed second waves. Genotype: *ss01613-Gal4 (Split3)/R94E10-lexA; lexAop-GCaMP6f,UAS-Chrimson:mCherry*. (**G**) MDNs are preferentially active during backward (BWD) not forward (FWD) locomotion in the intact larva. CaMPARI in MDN descending projections within the SEZ of third instar larvae. Top, fluorescence emission following excitation by 488 nm (green) or 561 nm (magenta); bottom, emission from 561 nm imaging alone. Right, quantification of red fluorescence over green fluorescence, mean intensity. Each value represents data from an individual descending projection. See Materials and methods for details. n = 22 for FWD and 19 for BWD. Scale bar, 10 µm. Genotype: *R49F02-Gal4^AD^ R53F07-Gal4^DBD^ UAS-CaMPARI*.
DOI: https://doi.org/10.7554/eLife.38554.006

405 nm illumination for 30 s while larvae moved either backward or forward. We detected little or no activity-induced red fluorescence during forward locomotion, but significant red fluorescence during backward locomotion (*Figure 2G*). We conclude that MDNs are active during backward but not forward locomotion.

## Identification of MDNs in a serial section TEM reconstruction of the larval CNS

To understand how MDNs induce backward locomotion, we next identified the MDN synaptic partners. To do this, we identified the MDNs in an existing serial section TEM reconstruction of the newly hatched larva (*Ohyama et al., 2015*). Our first step was to determine the precise morphology of both MDN neurons. We generated individually labeled neurons within the Split2 pattern using MultiColor FlpOut (MCFO) (*Nern et al., 2015*). These single neurons serve as the 'ground truth' for matching morphological features of individual neurons by light and electron microscopy (*Heckscher et al., 2015*; *Schneider-Mizell et al., 2016*). We identified single MDNs in Split2 MCFO preparations based on morphological similarity to the behavior flip-out neurons described in *Figure 2*. Diagnostic features shared by both MDNs in the pair include ventral, anterior, medial somata, distinctive ipsilateral and contralateral arbors, a contralateral projection in the posterior commissure, and descending projections terminating in segments A3-A5 of the VNC (*Figure 3A–E*). MDN descending projections run slightly lateral to the dorsal medial FasII^+^ bundle (*Landgraf et al., 2003*) (*Figure 3F*). Each neuron in the pair share all these features, but the two MDNs can be distinguished from each other by their ipsilateral arbor, which is either linear (*Figure 3C*, arrow) or bushy (*Figure 3D*, arrowhead). We next searched for the MDNs in the TEM volume using CATMAID (*Schneider-Mizell et al., 2016*). We found two pair of neurons that showed an excellent morphological match to the MDNs in every distinctive feature (*Figure 3A'–D'*); we annotate them as MDNa and MDNb in the TEM volume. Hereafter, we call these neurons simply MDNs due to their similarity in morphology and connectivity (see next section). Importantly, none of the 50 neurons with cell bodies nearest to the MDNs have a similar morphology (data not shown). Thus, we can be certain that the MDNs in the TEM reconstruction are identical to the MDNs visualized by our Split-gal4 lines. This is also confirmed by functional optogenetics (see below). We conclude that the MDNs can be uniquely identified by light microscopy and by TEM. Identification of the MDNs in the TEM volume is a prerequisite for identifying their pre- and post-synaptic partners (next section).

## The MDN circuit: three pathways to distinct premotor neurons

Annotation of the MDNs in the TEM reconstruction revealed bilateral arbors in the brain and descending processes to abdominal segments (*Figure 4A*). Pre-synapses are restricted to the descending processes (*Figure 4A*, green), whereas post-synapses are present in brain arbors and descending processes, suggesting information flow from brain to VNC. A representative MDN output synapse shown in *Figure 4B*; it is polyadic (multiple postsynaptic neurons clustered around the MDN pre-synapse) and electron dense with associated presynaptic vesicles.

Due to the ability of the MDNs to induce backward locomotion when activated, we focused on identifying MDN post-synaptic partners, with the goal of understanding the relationship between

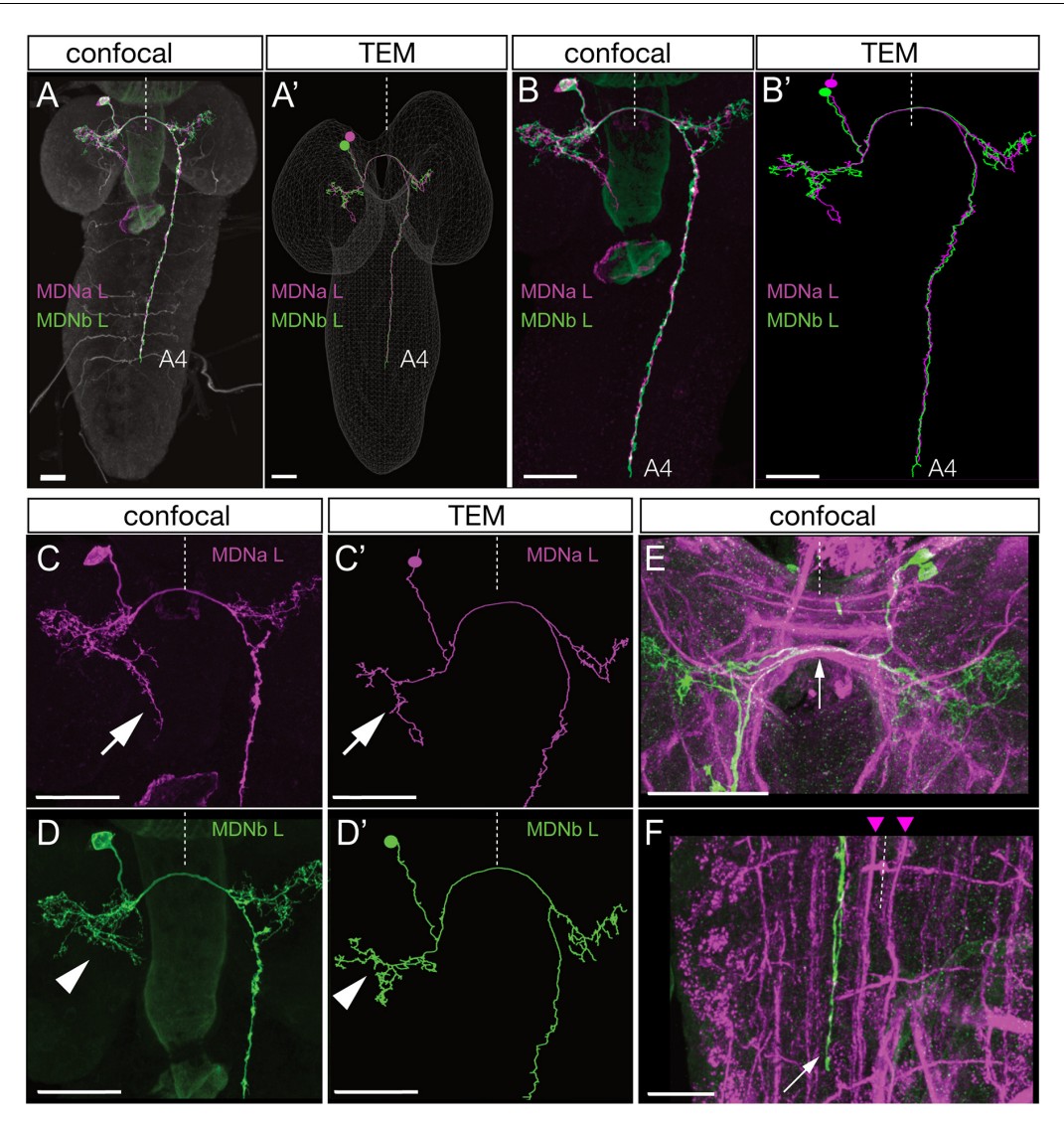

**Figure 3.** Identification of mooncrawler descending neurons by light and electron microscopy. (**A–F**) Light microscopy. Multicolor FLP-out (MCFO) was used to visualize the morphology of individual neurons in the Split2 pattern in first instar larvae. Two neurons show morphology matching that seen in the Chrimson FLP-out experiments in *Figure 2*. Both neurons have anterior medial somata (**A**), ipsilateral and contralateral arbors (**A–D**), a contralateral projection in the posterior commissure (**E**, arrow), and descending neurons terminating in segments A3-A5 of the VNC (**A–B**). The neurons run lateral to the dorso-medial (DM) FasII tract in the VNC (**F**, DM tract marked with arrowheads). The two neurons can be distinguished by their ipsilateral arbor, which is either linear (**C**, arrow) or bushy (**D**, arrowhead). (**A'–D'**) Reconstructions from serial section transmission electron microscopy (TEM) of a first instar larva. Two neurons indistinguishable from the MDNs can be identified in the TEM reconstruction: MDNa (linear ipsilateral arbor) and MDNb (bushy ipsilateral arbor). We simply call them MDNs due to their similar morphology and connectivity. All panels show dorsal views with midline indicated (dashed line). Scale bars, 20 μm.

DOI: https://doi.org/10.7554/eLife.38554.007

the MDN activation and motor output. The post-synaptic partners with the most synapses with MDN are: (1) the Pair1 SEZ descending neuron; (2) the thoracic descending neuron (ThDN); (3) the premotor neuron A18b; and (4) the MDNs themselves (*Figure 4C–D*). These are the top four MDN partners in both synapse number (*Figure 4C*) and percentage of total MDN output synapses (*Figure 4D*). All four MDNs have similar connectivity (*Figure 4—figure supplement 1*). Most of the top MDN output neurons are either premotor neurons or have preferential input into known premotor neurons (*Figure 4D–G*). For example, ThDN has a large number of synapses with A27l/A27k premotor neurons (*Figure 4C,E,H*), as well as with A18g (which is not a premotor neuron). Pair1 is connected to

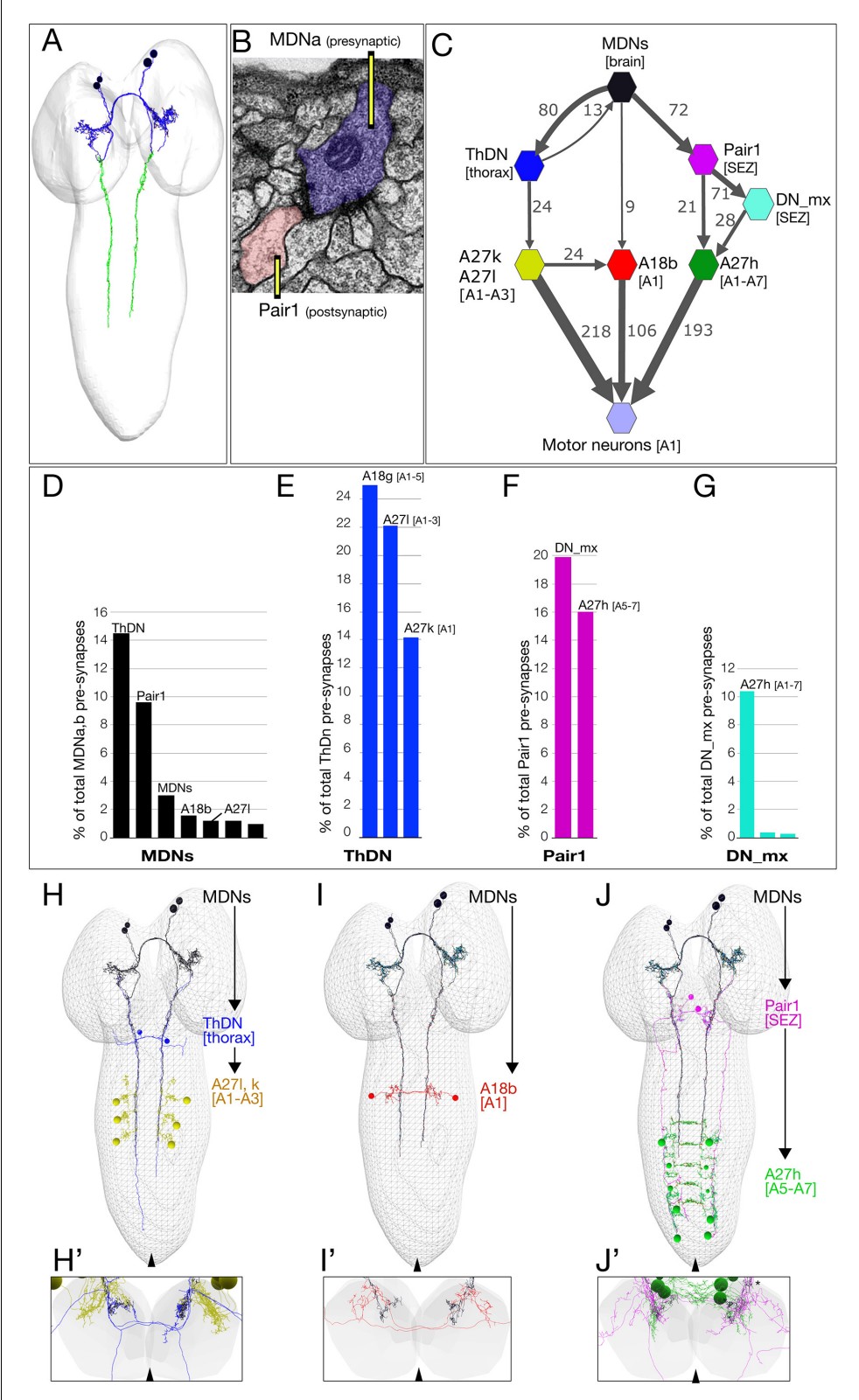

**Figure 4.** The MDN connectome: three pathways to distinct subsets of premotor neurons. (**A**) TEM reconstruction of the bilateral MDNa,b neurons. Neuronal skeletons are colored to show post-synapses in the presumptive dendritic arbors of the brain (blue) and pre-synapses in the presumptive axonal descending process (green). Anterior, up. (**B**) Representative MDN output pre-synapse (blue) onto a post-synaptic Pair1 neuron (pink). (**C**) MDNs and their partners with the greatest number of synapses (synapse number shown next to connection arrows, and line width is proportional to synapse

*Figure 4 continued on next page*

*Figure 4 continued*

number). All connectivities are shown except unilateral synapses,<6 synapses, and the 15 synapses between MDNs. Each polygon represents pairs of the indicated neuron with the exception of these larger groups: A27k/A27l (six A27l neurons in A1-A3, four A27l neurons in A1-A2); A27h (14 neurons in A1-A7), and 30 pair of motor neurons in A1. This graph is provided as *Supplementary file 1*. json that can be opened in CATMAID. (D–G) Quantification of the percent of total pre-synapses that are targeted to the indicated neuron. All connectivities are shown except unilateral or <5 synapse connections. (H–J) The three MDN to premotor neuron pathways. (H) MDN-ThDN-A27l/k pathway. Only A27l is shown; A27k has a very similar morphology. (I) MDN-A18b pathway. (J) MDN-Pair1-A27h pathway. Dorsal view; anterior, up; midline, arrowhead. (H′–I′) Respective cross-sectional view of VNC neuropil (gray) and neurons in each pathway; note that synapses are primarily in the dorsal (motor) neuropil. Dorsal up, midline, arrowhead. Asterisk in J′ shows the approximate site of the synapse shown in panel B.
DOI: https://doi.org/10.7554/eLife.38554.008

The following figure supplement is available for figure 4:

**Figure supplement 1.** All MDNs have similar connectivity.
DOI: https://doi.org/10.7554/eLife.38554.009

the previously described premotor neuron A27h (*Fushiki et al., 2016*), both directly and indirectly (*Figure 4C,F,J*). Lastly, A18b is a premotor neuron present in all abdominal segments, but it only receives MDN input in segment A1 (*Figure 4C,I*). Thus, the MDNs provide mono- and di-synaptic connectivity to premotor neurons. The activity and function of the MDN-A18b and MDN-Pair1 pathways in locomotion will be addressed below; we lack genetic tools to investigate the MDN-ThDN pathway (no known lines for ThDn or A27k, and the A27l line has many off-targets).

There are numerous MDN inputs (an average of 396 post-synapses per MDN neuron) and we have not attempted to reconstruct them; this is beyond the scope of a single paper. However, we note that each MDN has similar inputs. We do not detect mono-synaptic sensory input into the MDNs (data not shown), but based on the role of MDNs in generating a backward crawl in response to a noxious head touch, we predict that there will be, minimally, polysynaptic connections from head mechanoreceptors to the MDNs.

## MDNs activate A18b, a backward-active premotor neuron

The MDNs show anatomical connectivity to the A18b premotor neuron, which has not previously been characterized. We identified a LexA line that labels A18b within the VNC (R94E10, subsequently called A18b-LexA) along with a small, variable number of brain and thoracic neurons (*Figure 5—figure supplement 1*). A18b has local, contralateral projections that match the morphology of A18b in the TEM reconstruction (*Figure 5A*), is cholinergic (*Figure 5B*), and is connected directly to the dorsal-projecting motor neurons aCC/RP2 and U1/U2 (*Figure 5C*) among other motor neurons.

We showed above that MDNs are significantly more active during backward than forward locomotion, raising the question of whether the A18b neurons are also preferentially active during backward locomotion. To answer this question, we performed three experiments. First, we used dual color calcium indicators in a fictive CNS preparation to simultaneously monitor motor neuron activity (GCaMP6m) and A18b activity (jRCaMP1b). We observed robust forward and backward motor waves (*Figure 5D*, top), with A18b only active during backward motor waves, not forward motor waves (*Figure 5D*, bottom; quantified in *Figure 5E*). Second, we performed dual color calcium imaging within intact larvae, and again observed that A18b was only active during backward motor waves (*Figure 5D*; quantified in *Figure 5E*). Third, we used CaMPARI within intact larvae to determine if A18b was preferentially active during backward locomotion. We expressed CaMPARI in A18b and tested for activity-induced green-to-red photoconversion during either forward locomotion or backward locomotion. We found that illumination during forward locomotion generated minimal CaMPARI red fluorescence, whereas illumination during backward locomotion resulted in a significant increase in CaMPARI red fluorescence (*Figure 5F*). We call the A18b neuron backward-active rather than backward-specific because we do not know its pattern of activity in rolling or other larval behaviors. We conclude that A18b neurons are preferentially active during backward not forward locomotion.

To determine if MDNs activate A18b, we used Split1 to express Chrimson in MDNs and A18b-lexA to express GCaMP6f in A18b in fictive preparations. MDN stimulation led to a significant increase in GCaMP6f fluorescence in A18b, and this was not observed in controls lacking all-*trans*

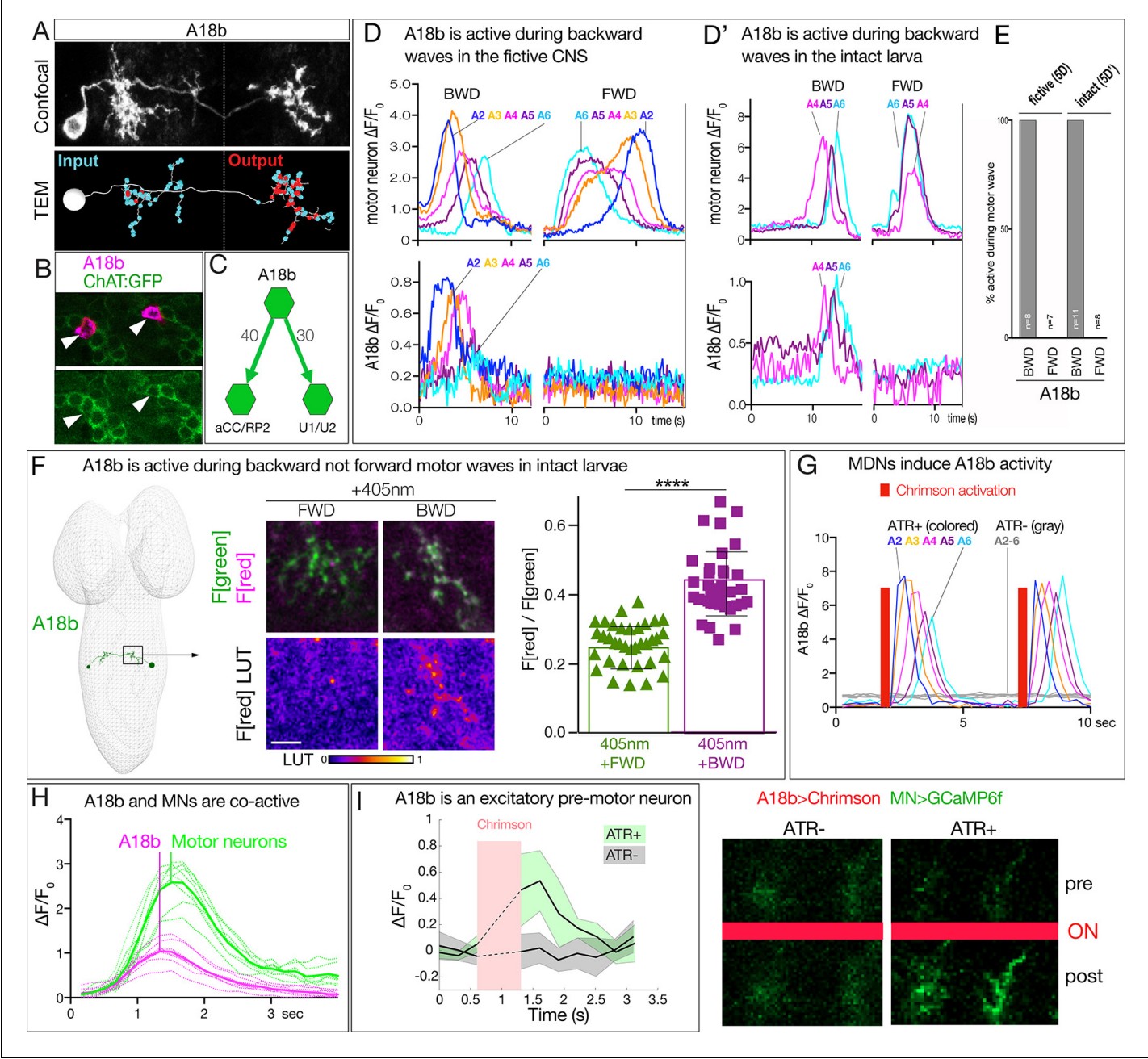

**Figure 5.** MDN activates the excitatory backward-active A18b premotor neuron. (**A**) A18b morphology by light (MCFO) and electron microscopy (TEM). Top: Dorsal view of an individual A18b neuron in a second instar larval CNS by light microscopy (R94E10 > MCFO). Bottom: Dorsal view of an individual A18b neuron in a first instar larva in the TEM reconstruction. Cyan dots, post-synaptic sites; red dots, pre-synaptic sites. Anterior, up. Midline, dashed line. Genotype: *R94E10-gal4 UAS-MCFO2*. (**B**) A18b is cholinergic. A18b cell body (mCherry; magenta) and ChAT:GFP (green). Genotype: *R94E10-Gal4, UAS-Chrimson:mCherry; mimic ChAT:GFP*. (**C**) Connectivity of A18b to neurons with the greatest number of A18b post-synapses: the dorsal-projecting motor neurons aCC/RP2 and U1/U2 in segment A1. Synapse number shown. (**D**) In fictive preparations, A18b neurons are active in backward but not forward locomotion. $\Delta F/F_0$ of GCaMP6m in U1-U5 motor neurons (top) or jRCaMP1b in A18b (bottom) of five segments executing a forward (FWD) and then a backward (BWD) wave. This experiment was performed on eight different isolated third instar CNSs with similar results; quantified in E. Genotype: *CQ-lexA/+; lexAop-GCaMP6m/R94E10-Gal4 UAS-jRCaMP1b.*. (**D'**) In intact larvae, A18b neurons are active in backward but not forward locomotion. $\Delta F/F_0$ of GCaMP6m in motor neurons (top) and jRCaMP1b in A18b (bottom) in three segments. Times of BWD and FWD motor waves indicated. This experiment was performed on 19 waves (11 BWD, 8 FWD) in seven third instar larvae, all with similar results; quantified in E. Genotype: *CQ-lexA/+; lexAop-GCaMP6m/R94E10-Gal4 UAS-jRCaMP1b.*. (**E**) Quantification of data in panels D and E. BWD, backward waves; FWD, forward waves. (**F**) In intact larvae, A18b is preferentially active during backward not forward locomotion. CaMPARI in A18b neurites in a third instar larval CNS. Top, fluorescence emission (**F**) following 488 nm (green) or 561 nm (magenta) illumination; bottom, emission from 561 nm alone. Left, photoconversion

*Figure 5 continued on next page*

*Figure 5 continued*

(405 nm) during FWD or BWD locomotion. Right, quantification of red fluorescence over green fluorescence mean intensity. Each value represents data from an individual neurite. n = 35 for FWD and 36 for BWD. LUT, 561 nm emission intensity look up table. Scale bar, 10 μm. Genotype: *R94E10-Gal4 UAS-CaMPARI*. (G) In fictive preparations, MDNs activate A18b neurons, and induce backward A18b activity waves. Chrimson is expressed in MDN, and GCaMP6f in A18b. Red bars, time of 561 nm Chrimson activation. Colored traces indicate the $\Delta F/F_0$ of A18b GCaMP6f signal in 5 segments of an ATR +brain; gray traces are from ATR- animal. This experiment was performed on five different animals with similar results. Genotype: *R49F02-Gal4^{AD}/ R94E10-lexA; R53F07-Gal4^{DBD}/lexAop-GCaMP6f UAS-Chrimson:mCherry*. (H) Dual color calcium imaging of jRCaMP1b in A18b (magenta) and GCaMP6m in U1-U5 motor neurons (green). In fictive preparations, A18b and motor neurons are co-active during backward waves. Both show similar initiation of activity, but A18b peak activity precedes motor neuron peak activity (vertical lines). Data are acquired every 168 ms from eight A18b/motor neuron pairs from three animals; peak activity of the motor neurons followed that of A18b by 0 ms (two pair), 168 ms (four pair), or 336 ms (two pair). Dashed lines, individual neurons; solid lines, average. Genotype: *CQ-lexA/+; lexAop-GCaMP6m/R94E10-Gal4 UAS-jRCaMP1b.'*. (I) A18b is an excitatory pre-motor neuron. A18b expresses Chrimson and aCC/RP2 motor neurons express GCaMP6f. Left: $\Delta F/F_0$ traces of GCaMP6f before and after 561 nm Chrimson activation (red bar) of three aCC/RP2 axons/dendrites within an animal. Solid bars represent means and shaded regions represent standard deviation from the mean (SDM). ATR +is shaded in green and ATR- in grey. Five animals were used in each group. GCaMP6f signal was not acquired during the Chrimson activation (dashed lines); t-test analysis for the first $\Delta F/F_0$ value after Chrimson activation between +ATR and -ATR showed significance (p=0.0071). Right: images of motor neuron GCaMP6f fluorescence pre- and post-Chrimson activation in ATR +and ATR- larvae. Genotype: *94E10-lexA/+; lexAop-Chrimson:mCherry/RRa-Gal4 UAS-GCaMP6f.*.

DOI: https://doi.org/10.7554/eLife.38554.010
The following figure supplement is available for figure 5:

**Figure supplement 1.** Driver lines for A18b.
DOI: https://doi.org/10.7554/eLife.38554.011

retinal (ATR), an essential co-factor for Chrimson function (*Figure 5G*). Interestingly, MDN activation triggered a backward wave of A18b activity from A2 to A6 (*Figure 5G*). We propose that MDN activates A18b in segment A1, which is the only segment we detect direct synaptic contacts, and this is transformed into an anterior-to-posterior wave of A18b activity.

We showed above that A18b has direct synaptic connectivity to motor neurons and is cholinergic, indicating that is likely to be an excitatory pre-motor neuron. Consistent with this expectation, we observed co-activity of A18b and motor neurons during backward motor waves in fictive preparation (*Figure 5H*), and A18b stimulation led to a significant increase in GCaMP6f fluorescence in motor neurons, which was not observed in controls lacking ATR (*Figure 5I*).

We wanted to test whether activation of A18b in segment A1 could induce backward waves of motor neuron activity. Unfortunately, the A18b-Gal4 line is not expressed in A1 (only A2-A7), precluding this experiment; moreover, it has 'off-target' expression in the brain and in the VNC; these off-target neurons do not prevent monitoring A18b activity because they do not overlap with A18b arbors, but they make it impossible to selectively activate or silence A18b. In conclusion, our data support the following model: MDN activates A18b in segment A1, which initiates a coordinated anterior-to-posterior wave of A18b/motor neuron activity that drives backward locomotion.

## MDNs activate Pair1, a backward-active descending interneuron

Connectomic data shows that MDNs have many synapses with the bilateral Pair1 neurons, which send a descending projection to the VNC where they form synapses with A27h in posterior abdominal segments. A27h neurons are only active during forward locomotion (*Fushiki et al., 2013*). This leads to the hypothesis we test below: MDNs activate Pair1 to inhibit A27h, which terminates forward locomotion.

To determine if MDNs activate Pair1 we used Split1 to express Chrimson in MDNs, and R75C02-lexA (hereafter Pair1-lexA) (*Figure 6A*) to express GCaMP6f specifically in Pair1. Stimulation of MDNs led to a significant increase in Pair1 GCaMP6f fluorescence, and this was not observed in controls lacking ATR (*Figure 6B*). We conclude that the MDNs activate Pair1 neurons. In addition, we observed that every time MDNs were active, the Pair1 neurons were co-active (n = 5; *Figure 6C*), although Pair1 could be active alone (n = 5; *Figure 6—figure supplement 1*). We conclude that MDNs activate the Pair1 neurons, and that other mechanisms exist for activating Pair1 as well (see Discussion).

We next used two methods to determine whether Pair1 neurons are preferentially active during backward locomotion. First, we used GCaMP6m to simultaneously monitor Pair1 and motor neuron activity in a fictive CNS preparation; this is possible because Pair1 and motor neuron processes are

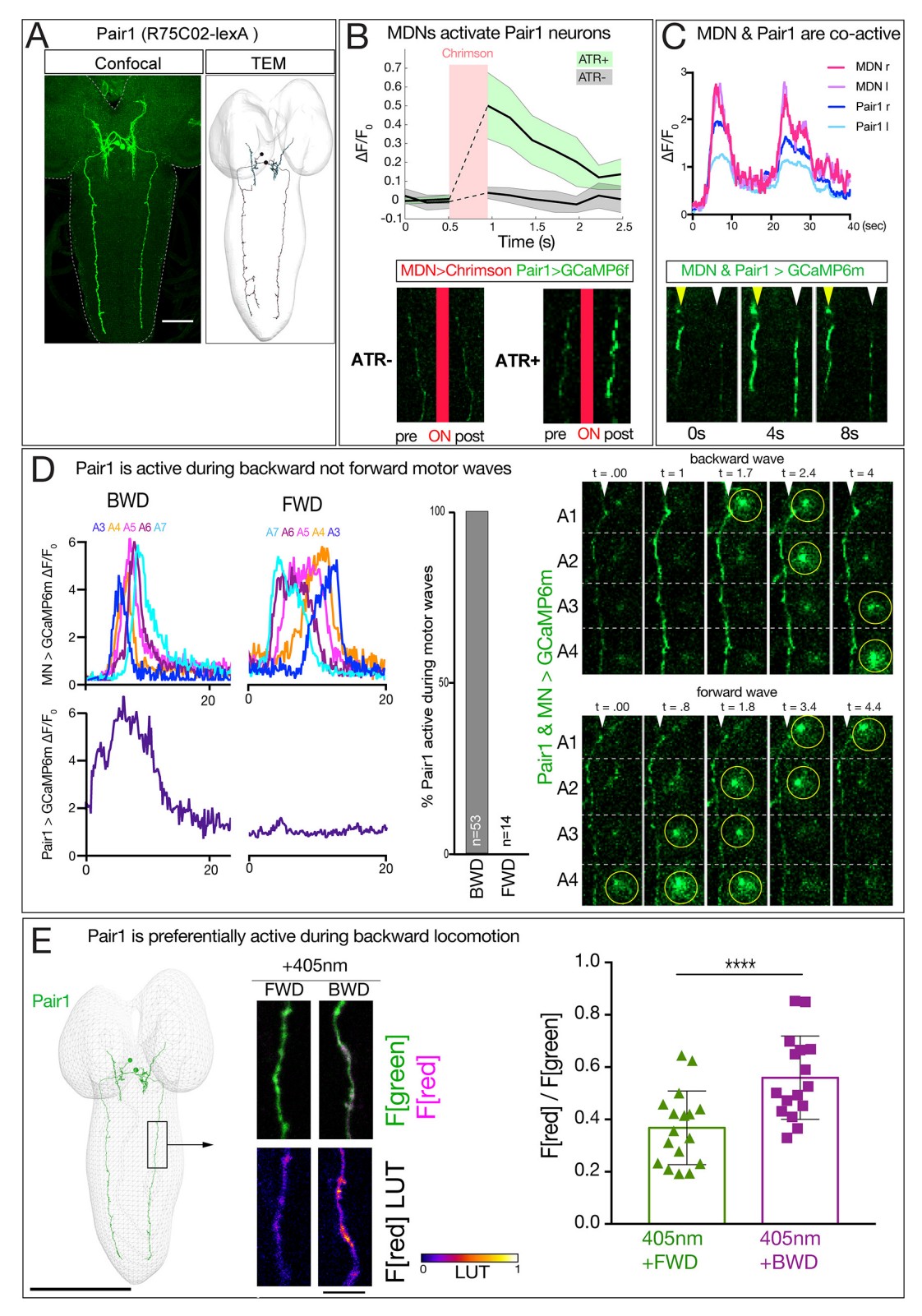

**Figure 6.** MDN activates Pair1 which is a backward-active descending neuron. (**A**) Pair1 neurons by light (confocal) and electron microscopy (TEM). Confocal image is an L3 CNS, TEM reconstruction is from an L1 CNS. Anterior, up. Scale bar, 50 μm. Genotype: *R75C02-lexA lexAop-myr:GFP..* (**B**) MDNs activate Pair1. MDN expresses Chrimson and Pair1 neurons express GCaMP6f. Top: ΔF/F$_0$ traces of GCaMP6f before and after Chrimson activation (red bar) of Pair1 axons. Solid bars represent means and shaded regions represent standard deviation from the mean. ATR +is shaded in

*Figure 6 continued*

green and ATR- in grey. Six animals were used for ATR +and five for ATR-. GCaMP6f signal was not acquired during the Chrimson activation (dashed lines); t-test analysis for the first $\Delta F/F_0$ value after Chrimson activation between +ATR and -ATR showed significance (p=0.0004). Bottom: images of Pair1 GCaMP6f fluorescence pre- and post-Chrimson activation in ATR +and ATR- larvae. Genotype: *R49F02-Gal4^AD^/R75 C02-lexA; R53F07-Gal4^DBD^/lexAop-GCaMP6f UAS-Chrimson:mCherry.* (**C**) MDN and Pair one are co-active. Top: MDN and Pair1 expressing GCaMP6m in different regions of the neuropil, and show concurrent activity. Bottom: MDNs (white arrowhead) and Pair1 (yellow arrowhead) show similar timing of GCaMP6m fluorescence during a BWD wave. Anterior, up; midline, right side of panel. MDN and Pair1 co-activity was observed in 5 out of 10 brains examined; the other five brains showed Pair1 activity but no MDN activity (see *Figure 6—figure supplement 1*). Genotype: *ss01613-Gal4/UAS-GCaMP6m; R75C02-Gal4.* (**D**) Pair1 is active during backward (BWD) but not forward (FWD) waves in fictive preparations. Left: Pair1 GCaMP6m activity (bottom) and motor neuron activity (top) during fictive BWD and FWD waves in the same animal. Pair1 is not active during FWD waves. Center: quantification. N = 53 BWD waves from seven different animals, and 14 FWD waves from four different animals. Right: Pair1 GCaMP6m activity (arrowheads) precedes U1-U5 motor neuron activity (circled). Genotype: *CQ-lexA/UAS-GCaMP6m; lexAop-GCaMP6m/R94E10-Gal4.* (**E**) Pair1 is preferentially active during backward locomotion in the intact animal. CaMPARI was expressed in Pair1 and photoconversion was activated during FWD or BWD locomotion in intact third instar larvae. There is significantly more CaMPARI photoconversion during BWD locomotion. Graph, quantification of red fluorescence over green fluorescence mean intensity. Triangle or square, data from an individual axon. n = 36 for FWD and 34 for BWD. Scale bar, 10 μm. Genotype: *R75C02-Gal4 UAS-CaMPARI..*

DOI: https://doi.org/10.7554/eLife.38554.012

The following figure supplement is available for figure 6:

**Figure supplement 1.** Pair1 can be activated independent of MDN activity.

DOI: https://doi.org/10.7554/eLife.38554.013

in different positions within the neuropil. These preparations show rhythmic forward and backward waves of motor neuron activity, and Pair1 neurons were only active during backward waves (*Figure 6D*; left, center). In cases where Pair1 activity is coupled with motor neuron activity, we find that Pair1 activity precedes motor neuron activity (*Figure 6D*, right). Second, we expressed CaM-PARI in Pair1 neurons and performed photoconversion during forward locomotion or backward locomotion. We found that illumination during forward locomotion generated a small amount of red fluorescence, whereas illumination during backward locomotion resulted in a significant increase in red fluorescence (*Figure 6E*). Taking all our anatomical and functional data together, we conclude that MDNs activate the A18b and the Pair1 neurons, which are both active during backward but not forward locomotion.

## Pair1 inhibits the A27h premotor neuron, arrests forward locomotion, and facilitates MDN-mediated backward locomotion

We confirm previous work (*Fushiki et al., 2016*) showing that A27h is active during forward not backward locomotion (*Figure 7—figure supplement 1*). This raises the interesting possibility that the MDNs coordinately switch locomotor behavioral states: concurrently promoting backward loco-motion via A18b, and suppressing forward locomotion via Pair1 inhibition of A27h.

To test whether Pair1 inhibits the A27h neuron, we expressed Chrimson in Pair1 and GCaMP6m in A27h. We used Chrimson to stimulate Pair1 just as A27h activity was rising as part of a forward motor wave, and observed a significant decrease in A27h GCaMP6m fluorescence; this was not observed in controls lacking ATR (*Figure 7A,B*). Furthermore, we found that Pair1 neurons are GABAergic (*Figure 7A''*), consistent with Pair1 direct repression of A27h activity. In addition, we found that Chrimson stimulation of Pair1 immediately and persistently blocked forward larval loco-motion; control larvae lacking ATR briefly paused in response to illumination onset but rapidly resumed forward locomotion (*Figure 7C*; *Videos 3* and *4*). Consistent with an inhibitory relationship, we observed that Pair1 and A27h activity is anti-correlated, with A27h often rising in activity as Pair1 declines in activity (*Figure 7—figure supplement 1*). We conclude that activation of the GABAergic Pair1 neurons inhibit A27h and prevent forward locomotion.

Our results suggest that Pair1 suppression of forward locomotion may be an essential component of MDN triggering a switch from forward to backward locomotion. If so, silencing Pair1 activity should reduce the effectiveness of MDN-induced backward locomotion; alternatively, MDN may be able to induce backward locomotion equally well without Pair1 function. Thus, we expressed Chrim-son in MDNs and the neuronal silencer Shibire^ts in Pair1; Shibire^ts blocks vesicle release at 32°C but not at 25°C (experiment summarized in *Figure 7D*). We observed that silencing Pair1 alone had no effect on forward locomotion (*Figure 7E,i–ii*), but silencing Pair1 prior to low light or high light

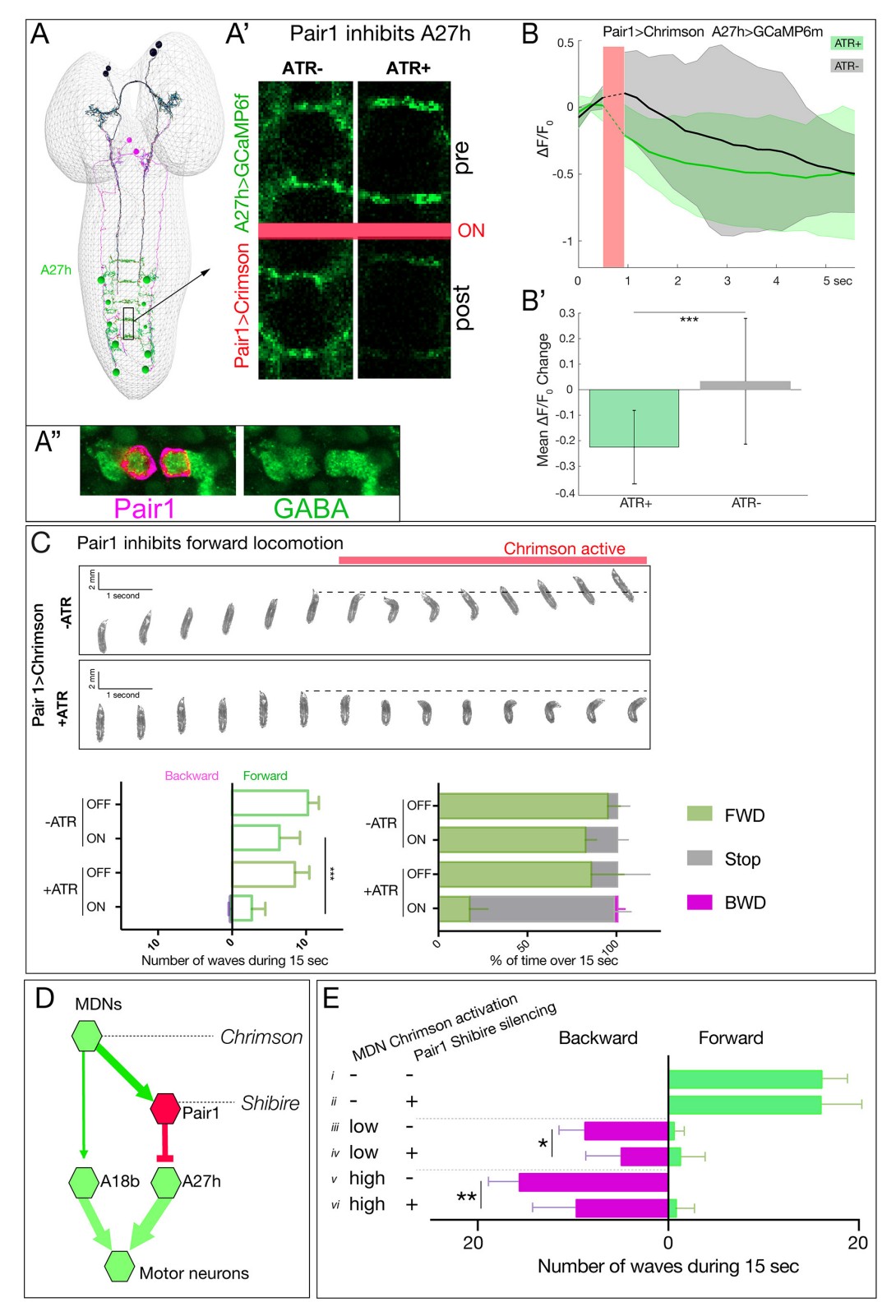

**Figure 7.** Pair1 inhibits the forward-active A27h premotor neuron, and arrests forward locomotion. (**A,B**) Pair1 inhibits A27h. (**A**) Reconstruction of MDNs (black), Pair1 neurons (magenta) and A27h neurons (green) in the first instar CNS TEM volume. (**A'**) A27h GCaMP6m fluorescence is reduced following Pair1 Chrimson activation (red bar); two segments shown. (**A''**) Pair1 is GABAergic. Pair1 cell body (mCherry; magenta, arrowheads) and GABA (green). Genotype: *R75C02-Gal4, UAS-Chrimson:mCherry*. (**B**) A27h GCaMP6m fluorescence is reduced following Pair1 Chrimson activation (red bar).

*Figure 7 continued on next page*

*Figure 7 continued*

(**B′**) ΔF/F$_0$ was significantly inhibited in ATR +animals relative to ATR- controls. A total of 26 events from seven animals were averaged for ATR +and 16 events from four animals for ATR- group. See Materials and methods for further details. Genotype: *R75C02-lexA/+; lexAop-Chrimson:mCherry/R36 G02-Gal4, UAS-GCaMP6m.*. (**C**) Activation of Pair1 halts FWD locomotion for the duration of neuronal activation. Top, time-lapse images of ±ATR larvae expressing Chrimson in Pair1 neurons before and during light stimulation. Bottom left, backward and forward wave number over 15 s without Chrimson activation (Off) or during Chrimson activation (On) in third instar larvae. n = 12 for all groups. Bottom right, percent time performing forward locomotion (green), backward locomotion (magenta) or not moving (grey) over 15 s without Chrimson activation (Off) or during Chrimson activation (On) in third instar larvae. n = 5 for all groups. Genotype: *R75C02-Gal4 UAS-Chrimson:mVenus.*. (**D**) Schematic illustrating the experiment in (**E**). Arrows, excitatory connections; T-bar, inhibitory connection; line width proportional to synapse number. (**E**) Pair1 activity is necessary for efficient Chrimson-induced backward locomotion. Chrimson was expressed in MDNs, and shibire$^{ts}$ was expressed in Pair1 neurons. Low (0.07 mW/mm$^2$) or high (0.275 mW/mm$^2$) light intensities were used to induce MDN activity; a temperature shift to 32°C was used to inactivate Shibire$^{ts}$ and thus silence Pair1 neurons. Silencing of Pair1 alone had no detectable phenotype (**i, ii**). Silencing Pair1 decreased the efficacy of MDN-induced backward locomotion at low or high light levels (iii-vi). Genotypes: *R49F02-Gal4$^{AD}$ R53F07-Gal4$^{DBD}$ UAS-Chrimson:mVenus pBD-lexA lexAop-Shibire$^{ts}$* (*i, iii* and *v*) and *R49F02-Gal4$^{AD}$ R53F07-Gal4$^{DBD}$ UAS-Chrimson:mVenus R75C02-lexA lexAop-Shi$^{ts1}$* (*ii, iv* and *vi*).
DOI: https://doi.org/10.7554/eLife.38554.014
The following figure supplement is available for figure 7:

**Figure supplement 1.** Timing of A27h neuronal activity.
DOI: https://doi.org/10.7554/eLife.38554.015

Chrimson-induced activation of MDN led to a loss in the effectiveness of MDN-induced backward locomotion (*Figure 7E,iii–vi*). We conclude that MDN triggers robust backward locomotion by coordinately activating the backward locomotion program and suppressing the forward locomotion program; we find no evidence for direct, monosynaptic reciprocal inhibition between these pathways (*Figure 7—figure supplement 1*).

## MDNs persist through metamorphosis and induce backward walking in adults

Larval MDNs share several features with the moonwalker descending neurons characterized in the adult (*Bidaye et al., 2014*; *Sen et al., 2017*). Both larval and adult neurons have anterior, medial somata with ipsilateral and contralateral arbors, and descending projections into the VNC. Both have presynaptic output into the SEZ and VNC. Could they be the same neurons? We tried to trace the MDNs through pupal stages using the Split1-Gal4 and observed the MDNs at early pupal stages

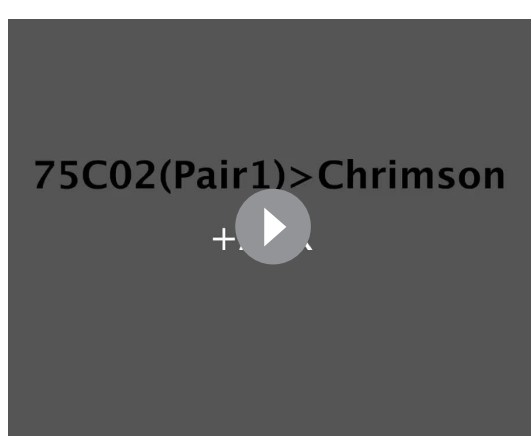

**Video 3.** Pair1 activation blocks forward locomotion. Crawling behavior of third instar larvae expressing Chrimson in Pair1 (75C02 > Chrimson:mVenus) with ATR. During the first 10 s, the animals are not under optogenetic light followed by 10 s under 0.28 mW/mm$^2$ of green light.
DOI: https://doi.org/10.7554/eLife.38554.016

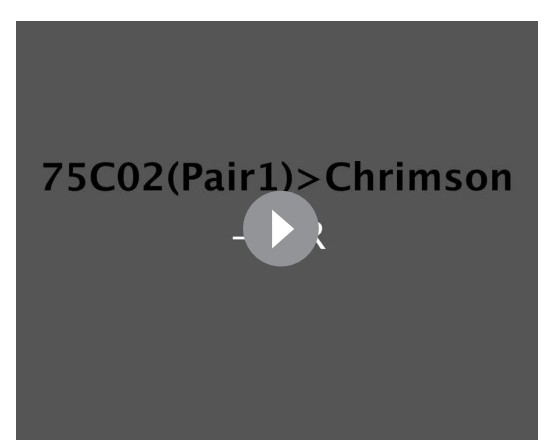

**Video 4.** Pair1 activation blocks forward locomotion. Crawling behavior of third instar larvae expressing Chrimson in Pair1 (75C02 > Chrimson:mVenus) without ATR. During the first 10 s, the animals are not under optogenetic light followed by 10 s under 0.28 mW/mm$^2$ of green light.
DOI: https://doi.org/10.7554/eLife.38554.017

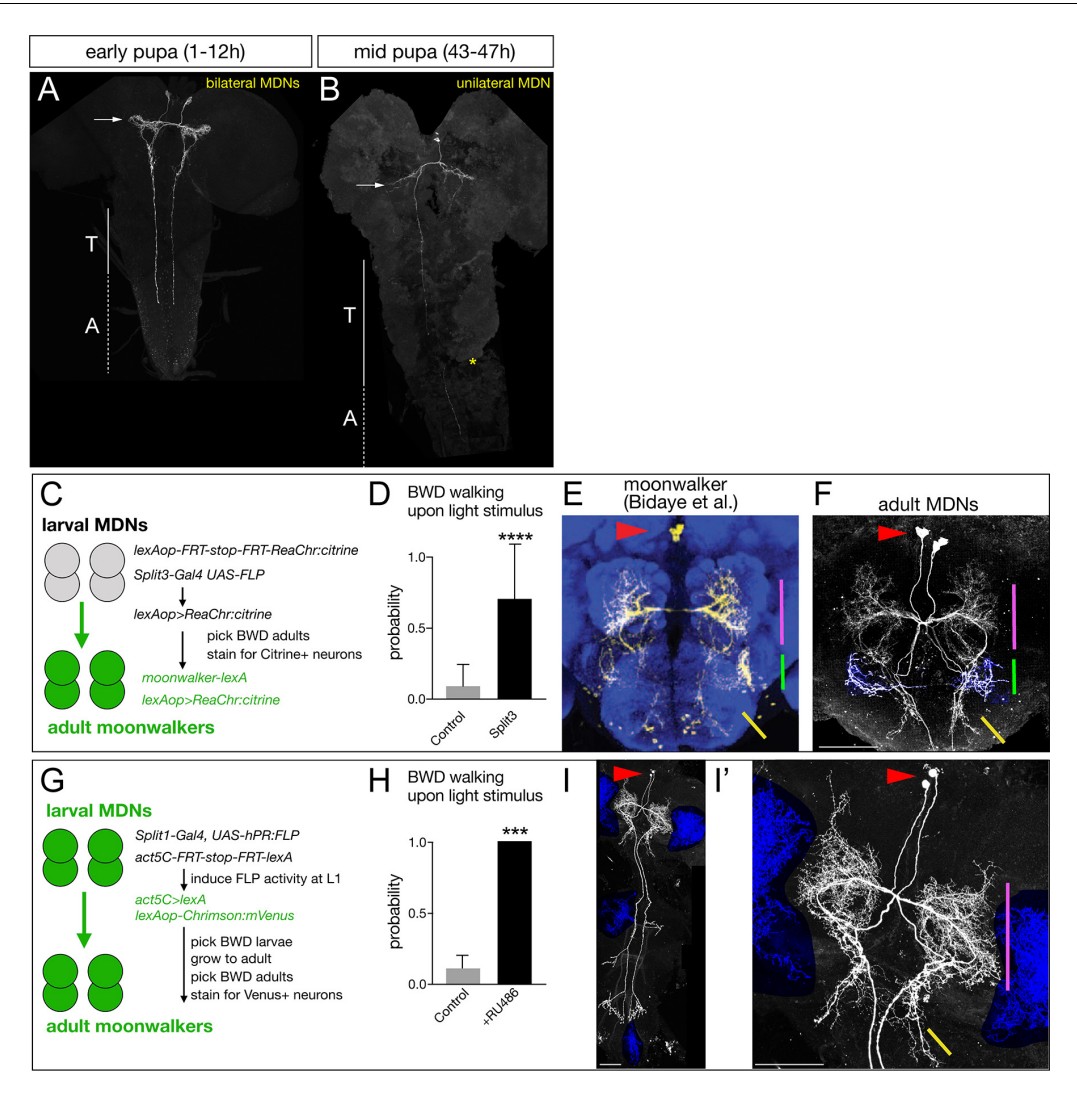

**Figure 8.** Larval MDNs persist into adulthood, match the moonwalker neuron morphology, and induce backward walking. (**A–B**) MDN neurons labeled by Split1 MCFO are similar in morphology to larval neurons during early pupal stages (**A**), but prune their brain and SEZ arbors by mid-pupal stages (B, arrow). T, thoracic segments; A, abdominal segments. Asterisk, tissue damage from dissection. (**C–F**) Larval MDNs persist to adulthood, match adult moonwalker morphology, and can induce backward walking in adults. (**C**) Genetic scheme for the experiment. Note that Split3 has no adult central brain expression (data not shown), and thus only the Split3 larval neurons will have the 'flipped out' lexAOP-ReaChr:citrine transgene. (**D**) Probability of adult backward walking upon light activation of Split3 immortalized neurons (split3) or controls lacking the DBD half of Split3 genotype (control). (**E**) Adult moonwalker neurons from *Bidaye et al., 2014*. Red arrowhead, cell bodies; colored lines, distinctive arbors. (**F**) One example of 'immortalized' larval MDNs showing the same cell body location (red arrowhead) and same distinctive arbors (colored lines); the arbor marked by the green line is an off-target projection not connected to the MDN neurons. Genotypes: Control: *UAS-FLP.PEST ss01613-(AD)-Gal4/TM3 VT044845-lexA lexAop-FRT-stop-FRT-ReaChr:citrine*. Split3: *UAS-FLP.PEST ss01613-(AD + DBD)-Gal4 VT044845-lexA lexAop-FRT-stop-FRT-ReaChr:citrine.*. (**G–I**) Larval MDNs persist to adulthood and induce backward walking. (**G**) Intersectional genetics used in this experiment. (**H**) Probability of adult backward walking upon light activation of the neurons immortalized by RU486-induced Flp activity (RU486+) or controls not given RU486 and thus lacking Chrimson expression in adult MDNs (control). (**I**) One example of an adult CNS plus VNC showing two MDNs (arrowhead) and four off target neurons (blue shading). (**I'**) Enlargement of brain showing MDNs and parts of two off target neurons (blue shading). Red arrowhead, cell bodies; colored lines, distinctive arbors in the protocerebrum (magenta line) and SEZ (green line). Scale bars, 50 μm.

DOI: https://doi.org/10.7554/eLife.38554.018

(*Figure 8A*) and mid-pupal stages, where they began to prune their dendritic arbors (*Figure 8B*). However, Split1 was down-regulated by adulthood (data not shown), requiring us to use alternate methods to follow the larval MDNs into adulthood.

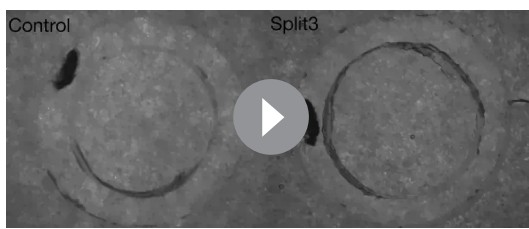

**Video 5.** Larval MDNs persist into adulthood and induce backward walking. Walking behavior of adult flies carrying all the components showed in 8F (split3, right) or all the genetic components except the DBD half of Split3 (control, left). During the first 10 s, the animals are not under optogenetic light followed by 10 s under 0.28 mW/mm² of red light.
DOI: https://doi.org/10.7554/eLife.38554.019

To permanently mark the larval MDN neurons, trace their morphology, and test their gain-of-function phenotype in the adult brain, we used two distinct intersectional genetic methods. First, we generated an intersection between a larval MDN line and an adult MDN line to express the optogenetic activator ReaChr (genetics schematized in *Figure 8C*). If the larval MDNs become adult moonwalker neurons, they will express ReaChr:citrine and show light-induced backward walking. We observed light-induced backward walking in 8 of 10 adult flies assayed (*Figure 8D*; *Video 5*); all eight had ReaChr::citrine expression in neurons matching the moonwalker neuron morphology (*Figure 8E,F*), whereas the two flies that did not walk backward also did not have ReaChr:citrine expression in moonwalker neurons (data not shown).

Second, we used 'immortalization' genetics (*Harris et al., 2015*) to permanently mark larval MDNs and assay their function in the larva and adult (genetics schematized in *Figure 8G*). We used Split1 to express an RU486-inducible FLP recombinase (hPR:FLP), allowing us to chemically induce FLP activity in first instar larva when Split1 is only expressed in the MDNs and a few off-targets. FLP activity resulted in permanent expression of lexA in the MDN neurons, which immortalizes expression of *LexAop-Chrimson:Venus* in these neurons. We identified larvae that crawled backward in response to Chrimson activation, and all grew into adults that showed Chrimson-induced backward walking (n = 20; *Figure 8H*). Importantly, all the backward walking adults that were successfully stained showed expression in the adult moonwalker neurons (n = 5; *Figure 8I,I'*); although each brain showed staining in a few additional neurons (blue shading), only the MDNs were present in all the brains.

We conclude that the larval MDNs are descending neurons that are born embryonically, persist throughout larval stages, and survive into the adult. Surprisingly, activation of MDNs can induce backward crawling in the limbless larva, as well as backward walking in the six-limbed adult (*Figure 9A*). How much of the MDN larval circuitry persists into the adult is an interesting open question (see Discussion).

## Discussion

We have shown that MDNs are brain descending interneurons that activate two neuronal pathways: one to stop forward locomotion and one to induce backward locomotion (*Figure 9B,C*). This is similar to *C. elegans*, where in response to a head poke the ASH sensory neuron activates AVA, a command neuron for backward locomotion (*Lindsay et al., 2011*), and indirectly inhibits AVB, a command neuron for forward locomotion (*Roberts et al., 2016*), although AVB inhibition may also arise from reciprocal inhibition between AVA and AVB. It is also similar to the role of the eighth nerve in simultaneously exciting the ipsilateral Mauthner neuron while inhibiting, via a feed-forward inhibitory neuron, the contralateral Mauthner neuron (*Koyama et al., 2016*). Our results raise the question of whether previously described command-like neurons in *Drosophila* (*Bidaye et al., 2014*; *King and Wyman, 1980*; *Sen et al., 2017*), leech (*Kristan, 2008*), lamprey (*Dubuc et al., 2008*), zebrafish (*Kimura et al., 2013*; *Medan and Preuss, 2014*), mouse (*Bouvier et al., 2015*; *Grillner and El Manira, 2015*; *Hägglund et al., 2010*; *Jordan et al., 2008*; *Juvin et al., 2016*; *Roberts et al., 2008*) and other animals may not only induce a specific behavior, but concurrently inhibit an antagonistic or incompatible behavior.

MDNs can induce backward locomotion within intact larvae or isolated CNS. In each case, a pulse of Chrimson activation as short as 300 ms can induce a full, multi-second long backward wave, suggesting that MDN activity triggers a backward wave without persisting throughout the wave. These results are consistent with detection of MDN-A18b synapses only in segment A1, and support our

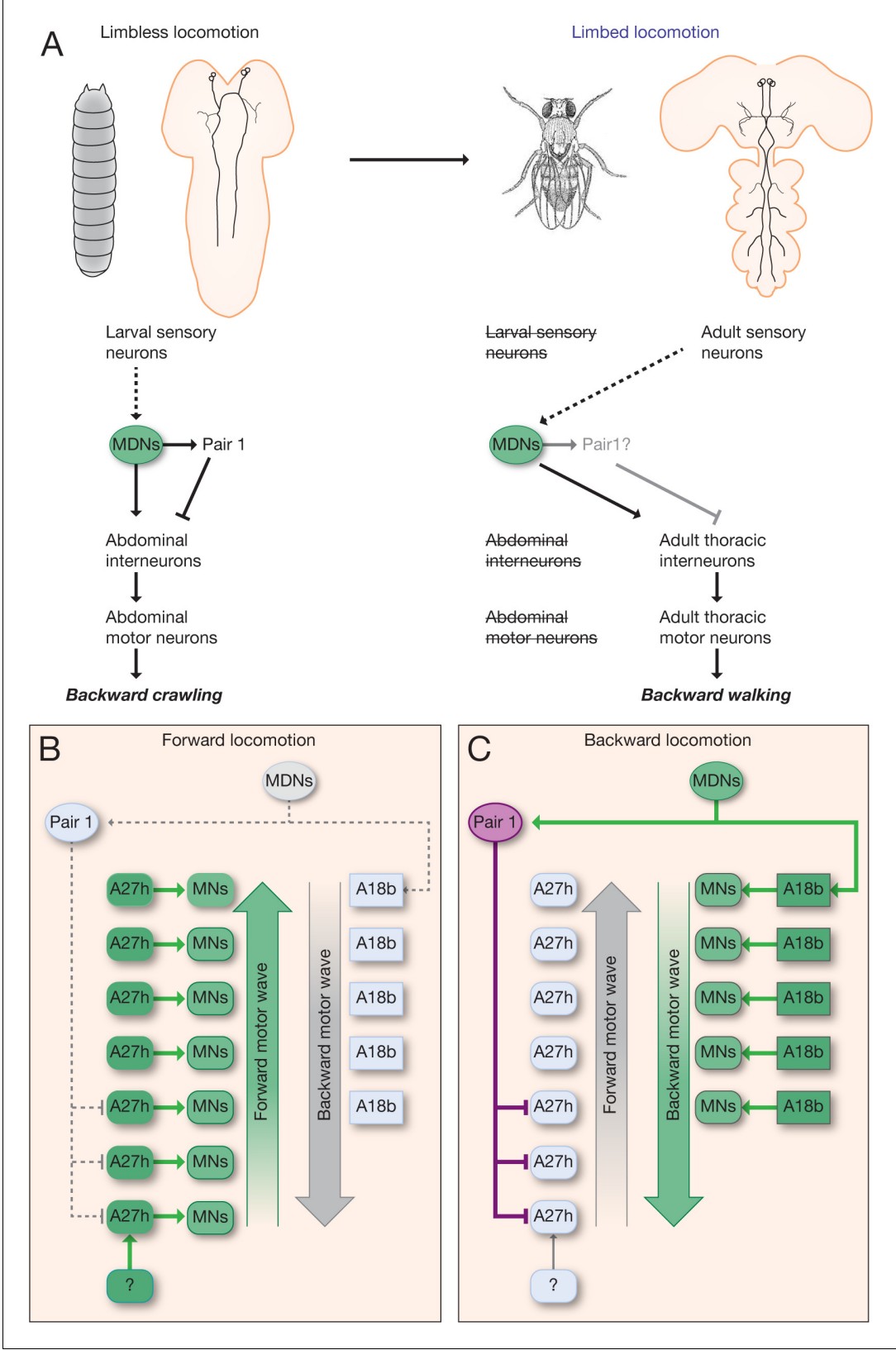

**Figure 9.** Model describing the MDN-mediated backward crawling. (**A**) MDN neurons are present in larval stages where they promote backward peristaltic crawling via abdominal premotor and motor neurons; MDNs subsequently persist into the adult fly where they promote backward walking of the six-limbed adult fly, using a

*Figure 9 continued on next page*

*Figure 9 continued*

different pool of thoracic motor neurons. Solid arrows represent direct excitatory connectivity; solid T-bars represent direct inhibitory connectivity; dashed arrows represent known (larva) or predicted (adult) polysynaptic connectivity. (B,C) Model. (B) During forward locomotion the MDNs, Pair1 and A18b are silent; an unknown neuron (?) may initiate forward locomotion. (C) To initiate backward locomotion, MDNs activate Pair1 descending neuron which inhibits the forward-active A27h premotor neuron in segments A5-A7 to halt forward locomotion. MDNs also activate A18b in A1 and/or in more anterior segments which triggers a backward motor wave.

DOI: https://doi.org/10.7554/eLife.38554.020

conclusion that MDNs trigger but do not persist throughout, a backward wave. Noxious stimuli that produce multiple waves of backward locomotion are likely to continuously activate MDNs.

We do not know how sensory stimulation of MDNs produce a bilateral backward motor wave. It may be that noxious sensory stimuli typically activate both left/right MDNs. Alternatively, there is a single contralateral synapse in both directions between A18b left/right neurons in A1, and the number could increase with the age of the larva. Perhaps, the single synapses between contralateral A18b neurons transform unilateral MDN activation into bilateral motor waves. Lastly, unilateral MDN activation may produce bilateral A18 activation via uncharacterized contralateral neurons.

MDNs are necessary for a normal backward locomotor response following mild noxious touch to the head. It is unclear how the tactile sensory cue is transduced to the MDNs: we find no monosynaptic sensory inputs to the MDNs in the current TEM connectome (data not shown). It is also unknown whether MDNs are used for backward crawling in response to other noxious sensory modalities, such as high salt, bright light, or bitter taste. MDNs may be dedicated to responding to noxious mechanosensation, or they may integrate multimodal inputs to initiate backward locomotion.

The discovery of MDN command-like neurons that switch locomotion from forward to backward raises the question: are there command-like neurons that induce the opposite transition: from backward to forward locomotion? Whereas the MDN descending projection extends to A3-A5, and thus well past the thoracic and upper abdominal segments that initiate backward locomotion (*Berni, 2015*; *Heckscher et al., 2012*; *Pulver et al., 2015*), a descending command-like neuron that induces forward locomotion is likely to project into the posterior abdominal segments, where forward waves are initiated (*Berni, 2015*; *Heckscher et al., 2012*; *Pulver et al., 2015*). Exploring the function of the latter type of descending neuron would help answer this question, as would the characterization of inhibitory inputs into the Pair1 or A18b backward-active neurons.

Our model is that the activation of A18b in A1 induces backward locomotion. This model is based on several observations. (1) A18b is only active during backward locomotion. (2) MDN forms excitatory synapses on A18b in A1 but not more posterior segments. (3) Stimulation of MDN produces an A18b backward activity wave. (4) The A18b backward wave is always concurrent with a motor neuron backward wave. Unfortunately, we are unable to directly test the function of A18b in triggering backward locomotion due to the A18b Gal4 line having off-target expression in the brain and in the VNC, and lacking expression in A1 or thoracic segments (*Figure 5—figure supplement 1*). Backward motor waves are initiated from the thorax (*Pulver et al., 2015*), and it is likely that stimulation of A18b in A1 or thoracic segments would be required to induce a backward motor wave. We attempted to find A18b in the thoracic segments, but failed, either due to incomplete annotation, segmental differences in morphology, or lack of thoracic A18b neurons. Similarly, the A02o 'wave' neuron can only induce backward motor waves following stimulation in anterior abdominal segments (*Takagi et al., 2017*). The relationship between A18b and A02o is unclear (they are not directly connected), nor is it known how activation of either produces a backward motor wave. This level of understanding would require a comprehensive anatomical and functional analysis of larval premotor and motor circuits.

We propose that MDNs directly excite Pair1 neurons to halt forward locomotion. But there are also additional mechanisms to induce Pair1 activity, as many Pair1 activity bouts occur without MDN activity. These alternate mechanisms are likely to be used for Pair1-induced pausing that is not followed by backward locomotion, for example during a pause-turn behavior. The MDN-independent inputs that activate Pair1 remain to be discovered.

The least understood MDN output to motor neurons is the MDN-ThDN-A27k/l pathway. A27l is inhibitory (AAZ and CQD, unpublished) so if ThDN is also inhibitory, it would provide a disinhibitory circuit motif for activating A18b. This would be synergistic with MDN direct excitation of A18b. There are currently no genetic tools providing access to ThDN or A27k neurons, and the existing driver line for A27l has off-target neurons, precluding a functional analysis of this pathway.

MDNs can induce backward crawling in the limbless *Drosophila* larva, and persist into adulthood where they can induce backward walking in the six-legged adult fly. This is remarkable because most mechanosensory neurons are completely different (*Kendroud et al., 2018*; *Kernan, 2007*), although there are some gustatory and stomatogastric sensory neurons that survive from larva to adult (*Kendroud et al., 2018*). Similarly, most or all the downstream motor neurons controlling crawling (larva) and walking (adult) are different: abdominal motor neurons in the larva and thoracic motor neurons in the adult. It will be interesting to see which, if any, interneurons in the larval MDN circuit remain connected in the adult, and whether they perform the same function in the adult. For example, does the larval Pair1-A27h circuit persist in the adult, but become restricted to thoracic segments? It is also interesting to consider the evolution of the MDN circuit; some of the neurons we describe here may originally have been used to regulate adult walking, prior to becoming co-opted for regulating larval crawling.

## Materials and methods

### Transgenes
*pBDP-Gal4* in attP2 (gift from B.D. Pfeiffer, JRC)
 *pBDP-LexA:p65Uw* in attp40 (gift from T. Shirangi, Villanova Univ)
 *R53F07-Gal4* (BDSC# 50442)
 *R53F07-Gal4$^{DBD}$* (Doe lab)
 *R49F02-Gal4$^{AD}$* (a gift from G. Rubin, JRC)
 *R94E10-Gal4* (A18b line; BDSC# 40689)
 *R94E10-lexA* (A18b line; Doe lab)
 *R36G02-Gal4* (A27h line; BDSC# 49939)
 *R75C02-Gal4* (Pair1 line; BDSC# 39886)
 *R75C02-lexA* (Pair1 line; a gift from M. Louis, UC Santa Barbara)
 *ss01613-Gal4* (Split3; a gift from M. Louis, UC Santa Barbara and J. Truman, Univ. Washington)
 *CQ2-lexA* (U1-U5 motor neurons; Doe lab)
 *RRa-Gal4* (aCC/RP2 motor neurons; a gift from M. Fujioka, Thomas Jefferson Univ.)
 *tsh-lexA* (a gift from J. Simpson, UC Santa Barbara)
 *UAS-Chrimson:mCherry* (a gift from V. Jayaraman, JRC)
 *UAS-Chrimson:mVenus* (BDSC# 55138)
 *UAS-dsFRT.Chrimson:mVenus* (a gift from G. Rubin, JRC)
 UAS-MCFO$_2$ (BDSC# 64086)
 *UAS-GCaMP6m* (BDSC# 42748)
 *UAS-GCaMP6f* (a gift from V. Jayaraman, JRC)
 *UAS-jRCaMP1b* (BDSC# 63793)
 *lexAop-GCaMP6f* (gift from V. Jayaraman, JRC)
 *lexAop-Gal80* (BDSC# 32213)
 *lexAop-Chrimson:mCherry* (a gift from V. Jayaraman, JRC)
 *lexAop-KZip+:3xHA* (a gift from B. White, NIH)
 *UAS-CaMPARI* (BDSC# 58761)
 *UAS-GtACR1* (a gift from A. Claridge-Chang, Duke-NUS Med School)
 *lexAop-shibire$^{ts}$* in attP2 (a gift from G. Rubin, JRC)
 *VT044845-lexA* (adult moonwalker line; a gift from B. Dickson, JRC)
 *hsFlpG5.PEST* (BDSC# 62118)
 *pJFRC108-20XUAS-IVS-hPR:Flp-p10* (a gift from J. Truman, Univ. Washington)
 *Actin5C-FRT>-dSTOP-FRT>-LexAp:65* (a gift from J. Truman, Univ. Washington)
 *P[13XLexAop2-IVS-CsChrimson.mVenus] attP18* (BDSC# 55137)
 *lexAop-(mCherry-STOP-FRT) ReaChR:Citrine VK00005* (BDSC #53744)

## Fly stocks

Split1 (*R53F07-Gal4^{DBD} R49F02-Gal4^{AD}*)

   Split2 (*R53F07-Gal4^{DBD} R49F02-Gal4^{AD} tsh-lexA lexAop-KZip+:3xHA*)

   Split3 (*ss01613-Gal4*)

   Immortalization stock: *P[13XLexAop2-IVS-CsChrimson.mVenus]attP18; Actin5C-FRT-STOP-FRT-lexAop::65; pJFRC108-20XUAS-IVS-hPR::Flp-p10*

## Immunostaining and imaging

Standard confocal microscopy, immunocytochemistry and MCFO methods were performed as previously described for larvae (*Clark et al., 2016*; *Heckscher et al., 2015*) or adults (*Nern et al., 2015*; *Pfeiffer et al., 2008*). Primary antibodies used recognize: GFP or Venus (rabbit, 1:500, Thermo-Fisher, Waltham, MA; chicken 1:1000, Abcam13970, Eugene, OR), GFP or Citrine (Camelid sdAB direct labeled with AbberiorStar635P, 1:1000, NanoTab Biotech., Gottingen, Germany), GABA (rabbit, 1:1000, Sigma, St. Louis, MO), mCherry (rabbit, 1:1000, Novus, Littleton, CO), Corazonin (rabbit, 1:2000, J. Veenstra, Univ Bordeaux), FasII (mouse, 1:100, Developmental Studies Hybridoma Bank, Iowa City, IA), HA (mouse, 1:200, Cell signaling, Danvers, MA), or V5 (rabbit, 1:400, Rockland, Atlanta, GA), Flag (rabbit, 1:200, Rockland, Atlanta, GA). Standard methods were used for pupal staging (*Bainbridge and Bownes, 1981*). Secondary antibodies were from Jackson Immunoresearch (West Grove, PA) and used according to manufacturer's instructions. Confocal image stacks were acquired on Zeiss 700, 710, or 800 microscopes. Images were processed in Fiji (https://imagej.net/Fiji), Adobe Photoshop (Adobe, San Jose, CA), and Adobe Illustrator (Adobe, San Jose, CA). When adjustments to brightness and contrast were needed, they were applied to the entire image uniformly. Mosaic images to show different focal planes were assembled in Fiji or Photoshop.

## Electron microscopy and CATMAID

We reconstructed neurons in CATMAID using a Google Chrome browser as previously described (*Ohyama et al., 2015*). Figures were generated using CATMAID graph or 3D widgets.

## Chrimson and GtACR behavioral experiments

Embryos were collected for 4 hr on standard 3.0% agar apple juice collection caps with a thin layer of wet yeast, and transferred to standard cornmeal fly food supplemented with 0.5 mM all-*trans* retinal at 48 hr after collection. Following another 48 hr (96 ± 6 hr larval age), animals were collected and transferred to 3.0% agar apple juice caps and relocated to the room were behavioral data was collected. Five minutes after acclimation to the room, one animal at a time was transferred to of 3.0% agar apple juice square arenas, 2 cm thick with an area of 81.0 cm$^2$, and crawling was then recorded at 5 Hz using an Axiocam 506 mono under low transmitted light from below for 15 s follow by 15 s under 0.275 mW/mm$^2$ 561 nm green light. For the intact larvae experiment in *Figure 2F*, a 300 ms pulse of 0.275 mW/mm$^2$ 561 nm green light was followed by 5 s of recording in the absence of green light. For the fictive CNS experiment in 2F, a 300 ms pulse of 561 nm green light was followed by 25 s of recording in the absence of green light. Temperature of the room was kept at 24 ± 2C°. Number of forward waves and backward waves, and percent of time engaged in either forward, backward or paused were quantified using the recorded movies. Behavioral data was acquired, given an unique identifier, and scored blind; except *Figure 7E*, where it was a binary assay (forward wave/backward wave) that did not require blind scoring. Unpaired Student's t-test was performed to determine significance in the number of waves over 15 s.

   The Chrimson together with Shibire silencing experiment (*Figure 7D–E*) was performed as the Chrimson only experiments described above except that the agar arena was placed on top of a heating plate which was kept at 25C° or at 32C° for Shibire Off or On groups respectively. Animals were individually placed on the arena. After 1 min to reach the desired temperature, we manually quantified the number of forward and backward waves with no light, under 0.07 mW/mm$^2$ green light or 0.275 mW/mm$^2$ green light.

   For GtACR1 experiments (*Figure 1G–H*), instead of square arenas, animals were placed into a 0.75 mm wide agar lane to limit their movement to forward or backward locomotion only. To quantify backward wave probability (*Figure 1G*) larvae were gently poked in the most anterior part of their body and scored whether the animal responded with backward crawling (regardless of how

many backward peristaltic waves). We then calculated the probability by dividing the number of times the animal began backward crawling immediately after a poke by the total number of times that each animal was poked, which was always five times. For each animal, this was done with no light first and then under 0.96 mW/mm$^2$ 561 nm green light. We performed one-way ANOVA with Bonferroni post-hoc test between light ON groups. For panel 1H, we induced a backward run and turned on the 0.96 mW/mm$^2$ green light immediately after the second backward wave. We define a backward run as two or more consecutive backward peristaltic waves after being poked in the most anterior part of the animal. We scored how many backward waves animals performed after the light was turned on.

## Calcium imaging

For dual-color and single-color calcium imaging in fictive preps, freshly dissected brains were mounted on 12 mm round Poly-D-Lysine Coverslips (Corning BioCoat) in HL3.1 saline, which were then were placed on 25 mm ×75 mm glass slides to be imaged with a 40 × objective on an upright Zeiss LSM-800 confocal microscopy. To do calcium imaging in intact animals (e.g. *Figure 5D′*), a second or third instar larva was washed with distilled water, then moved into a drop of halocarbon oil 700 (Sigma, St. Louis, MO) on the slide. A 22 mm × 40 mm cover glass was put on the larva and pressed gently to restrict larval locomotion. The larva was mounted ventral side up so that the ventral nerve cord could be imaged using 40 × objective on an upright Zeiss LSM800 confocal microscope. To simultaneously image two different neurons expressing GCaMP, we imaged neuron-specific regions of interest (ROI). In addition, we imaged two neurons using neuron-specific GCaMP6m and jRCaMP1b. Image data were imported into FijI (https://imagej.net/fiji) and GCaMP6m and jRCaMP1b channels were separated. The $\Delta F/F_0$ of each ROI was calculated as $(F-F_0)/F_0$, where $F_0$ was averaged over ~1 s immediately before the start of the forward or backward waves in each ROI.

## Functional connectivity assays

Freshly dissected brains were mounted in HL3.1 saline as described above, with the exception that the dissection was done under the minimum level of light possible to prevent activation of Chrimson. GCaMP6m or GCaMP6f signal in postsynaptic neurons were imaged using 2–4% power of the 488 nm laser with a 40 × objective on an upright Zeiss LSM800 confocal microscope. Chrimson in presynaptic neurons was activated with three pulses of 561 nm laser at 100% power delivered via the same 40 × objective using the bleaching function in the ZEN Zeiss software. The total length of the pulses carried was depend on the ROI size which was kept consistent across ATR +and ATR– samples within an experiment. For A18b activation (*Figure 5I*), the light pulse was 700 ms; for activation of MDN (*Figure 6B*) or Pair1 (*Figure 7B*), the light pulse was 440 ms. To quantify $\Delta F/F_0$ traces we used MAT-LAB. Before extracting any fluorescence, our script first performs rigid registration to correct for movement while recording. $F_0$ was set as the average fluorescence of the three frames acquired before each Chrimson stimulus analyzed. For predicted excitatory connections (*Figures 5I* and *6B*), we first average $\Delta F/F_0$ traces for two consecutive 561 nm Chrimson stimuli separated by 20 488 nm acquisition frames. This was enough time to let GCaMP6f levels return to ground state. For predicted inhibitory connections (*Figure 7B*), we gave multiple 440 msec Chrimson stimuli separated by 5 s. After recording, we then selected all events where the start of the Chrimson stimulus coincided with an A27h forward activity wave, which was necessary to elevate the GCaMP6m levels sufficiently to see subsequent Chrimson-induced inhibition. We selected the A27h segmental neuron with the highest mean fluorescent intensity in the frame before the Chrimson stimulus from segments A4-A7 (where Pair1 synapses with A27h). For all Chrimson experiments, traces were averaged across animals.

## CaMPARI experiments

Larvae were collected 96 hrs days after egg laying and place in agar apple collection caps for at least 5 min to acclimate animals to the environment. Using a soft brush, larvae were placed into a 0.75 mm wide agar lane to limit their movement to forward or backward locomotion only. We let the animals start crawling forward for at least 5 s in the lanes. For forward data collection, the photoconverting 405 nm light was turned on at 0.5 mW/mm$^2$ while the larvae crawled forward for 30 s. For

backward, same light was turned on and backward locomotion was immediately induced by gentle touch on the most anterior part of the larva with a semi-blunt pin. Brains were dissected in HL3.1, then green and red CaMPARI signals were imaged with a 40 × objective on Zeiss LSM-800 confocal microscope in the regions of interest. ROIs were manually selected using the green channel. Fluorescence within ROIs were quantified using Image J.

### Adult behavioral intersectional experiment (*Figure 8C,D,F*)

After eclosion adults were transferred to standard cornmeal fly food supplemented with ATR (0.5 mM) for 4 days changing to fresh food after two days. Wings were clipped and animals were placed in ring arenas made of 3.0% agar apple juice. The ring arena size was 1.4 cm outer diameter, 1.0 cm inner diameter and 0.2 cm height. After 5 min for environmental acclimation, animal behavior was recorded at 5 Hz using an Axiocam 506 mono under low transmitted light for 10 s followed by 10 s under 0.28 mW/mm$^2$ red light. This was done three times for each animal. To quantify backward locomotion probability upon light stimulus, we divided the amount of times the animal began backward walking within 2 s after light stimulus over the total number of times the animals was presented with light. To calculate significance, we used Student's t-test unpaired analysis.

### Adult behavior immortalization with RU486 experiment (*Figure 8G–I*)

Adult flies were allowed to lay eggs on standard culture medium that was supplemented with 1 μM RU486 and 2 mM ATR. After 24 hr, light-induced backward crawling larvae were transferred to culture medium supplemented with 2 mM ATR and grown to adulthood. Two- to 6-day-old adult flies were individually transferred into a 10-ml serological pipette for walking assay. Red-orange light from a 617 nm high-power LED was fiber-coupled to a 200 μm core optical cable that was triggered via a T-Cube LEDD1B driver (ThorLabs, Newton, NJ). Optogenetic stimulation was measured via a photodiode power sensor (S130VC, ThorLabs) to be ~4.6 μW/mm$^2$. We performed the same analysis for the intersectional experiment (above) to quantify backward locomotion probability upon light stimulus.

### Statistical analysis

Statistical significance is denoted by asterisks: ****$p < 0.0001$; ***$p < 0.001$; **$p < 0.01$; *$p < 0.05$; n.s., not significant. All statistical Student's t-tests were performed using Graphpad Prism software. One way ANOVA with Bonferroni post-hoc test was done using http://astatsa.com/. The results are stated as mean ± s.d., unless otherwise noted.

## Acknowledgements

We thank Jim Truman and Matthieu Louis for unpublished fly lines targeting MDN and Pair1 neurons. We thank Brandon Mark for MATLAB scripts, and Cooper Doe for brain dissections. We thank Todd Laverty, Gerry Rubin, Barry Dickson, Ben White, Miki Fujioka, Julie Simpson and Adam Claridge-Chang for fly stocks, and Barry Dickson, Shawn Lockery, Matthieu Louis, and Tory Herman for comments on the manuscript. We thank Avinash Kandelwal and Laura Herren for annotating neurons, and Keiko Hirono for generating transgenic constructs. Transgenic lines were generated by BestGene (Chino Hills, CA) or Genetivision (Houston, TX). Stocks obtained from the Bloomington Drosophila Stock Center (NIH P40OD018537) were used in this study. Funding was provided by HHMI (CQD, AC-R, AAZ, LM), NIH HD27056 (CQD, MQC), F32NS105350-01A1 (AC-R), APS Porter Physiology Development Fellowship, T32HD007348-24 (MQC), and T32GM007413-36 (MQC).

## Additional information

### Funding

| Funder | Grant reference number | Author |
| --- | --- | --- |
| National Institutes of Health | F32NS105350-01A1 | Arnaldo Carreira-Rosario |

| | | |
|---|---|---|
| Howard Hughes Medical Institute | HHMI | Arnaldo Carreira-Rosario<br>Aref Arzan Zarin<br>Laurina Manning<br>Chris Q Doe |
| National Institutes of Health | T32HD007348-24 | Matthew Q Clark |
| National Institutes of Health | HD27056 | Matthew Q Clark<br>Chris Q Doe |
| APS Porter Physiology Development Fellowship | T32HD007348-24 | Matthew Q Clark |
| APS Porter Physiology Development Fellowship | T32GM007413-36 | Matthew Q Clark |

The funders had no role in study design, data collection and interpretation, or the decision to submit the work for publication.

## Author contributions

Arnaldo Carreira-Rosario, Aref Arzan Zarin, Formal analysis, Validation, Investigation, Methodology, Writing—review and editing; Matthew Q Clark, Conceptualization, Data curation, Formal analysis, Investigation, Methodology, Writing—original draft, Writing—review and editing; Laurina Manning, Formal analysis, Investigation, Methodology, Writing—review and editing; Richard D Fetter, Albert Cardona, Writing—review and editing; Chris Q Doe, Conceptualization, Resources, Data curation, Supervision, Funding acquisition, Validation, Writing—original draft, Project administration, Writing—review and editing

## Author ORCIDs

Aref Arzan Zarin http://orcid.org/0000-0003-0484-3622
Matthew Q Clark https://orcid.org/0000-0002-1113-9388
Richard D Fetter http://orcid.org/0000-0002-1558-100X
Albert Cardona http://orcid.org/0000-0003-4941-6536
Chris Q Doe http://orcid.org/0000-0001-5980-8029

## Decision letter and Author response

Decision letter https://doi.org/10.7554/eLife.38554.024
Author response https://doi.org/10.7554/eLife.38554.025

# Additional files

## Supplementary files

• Supplementary file 1. Graph view of the MDN and downstream neurons. File can be opened in CATMAID using the graph widget.
DOI: https://doi.org/10.7554/eLife.38554.021

• Transparent reporting form
DOI: https://doi.org/10.7554/eLife.38554.022

## Data availability

All data presented in this study are available as supplementary files.

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
