## [Decision Letter]

Thank you for submitting your article "A command-like descending neuron that coordinately activates backward and inhibits forward locomotion" for consideration by *eLife*. Your article has been reviewed by three peer reviewers, including Ronald L Calabrese as the Reviewing Editor and Reviewer #1, and the evaluation has been overseen by Eve Marder as the Senior Editor.

The reviewers have discussed the reviews with one another and the Reviewing Editor has drafted this decision to help you prepare a revised submission.

Summary:

In this manuscript, the authors present a thorough analysis of descending neurons (2 bilateral pairs of MDNs) that can initiate backward crawling in *Drosophila* larvae; their role in behavior and their downstream circuitry that can mediate a transition from forward to backward crawling. They show that activating the neurons is sufficient to suppress forward crawling and initiate backward crawling. They demonstrate also that these neurons are active during backward crawls using CaMPARI albeit not in every case. They then show that suppressing their activity suppresses backward crawling in response to mild mechanical stimuli to the anterior body, indicating necessity for this specific behavior. They use immortalization genetics to follow these neurons into the adult where they have been previously described as 'command' neurons for backward walking. The manuscript uses the full armamentarium of intersectional genetics and optogentic techniques available for circuit analyses and is guided by an EM reconstruction of downstream synaptic targets. The level of circuit analyses is impressive and clearly shows that there is parallel excitation of backward crawling circuitry and inhibition of forward crawling circuitry. The analysis is elegant and pushes the technology available. The elegance of both the analysis and the mechanisms uncovered for preventing conflicting behaviors ensures that the manuscript will be of wide interest.

Essential revisions:

1a) The paper does not consider the dynamics of backward locomotion, and the role of the MDNs and downstream circuits in generating these dynamics. Reviewer #2 describes 2 experiments that could address this issue. The second experiment (varying the temporal properties of MDN optogenetic stimulation, and measuring the effects on behavior or downstream interneuron activity) seems doable and would definitely enhance the paper but it is not critical, and if it cannot be performed within the timeframe of revisions, this concern should be dealt with as a caveat in the Discussion.

1b) The statement that "…MDNs coordinate a transition between antagonistic larval locomotor behaviors…" seems an overreach unless some data is presented on the relationship between MDN activity and backward locomotion.

2a) Command is always a sticky issue because it is often loosely used. The authors provide a nice sufficiency test but perform a necessity test only for backward crawling in response to mild mechanical stimulation of the anterior body and suppression is not 100% complete. We suggest that the authors clearly define command, if they wish to maintain this terminology and that they carefully circumscribe its usage.

2b) Please clarify why the MDNS are not always active during backward crawls as assessed by CaMPARI.

*Reviewer #2:*

1) The paper does not consider the dynamics of backward locomotion, and the role of the MDNs and downstream circuits in generating these dynamics. For example, is MDN activity sustained throughout backward locomotion? Is transient activation of the MDNs sufficient to produce sustained backward movement? At the circuit level, the first question could be addressed by imaging from MDN neurons during fictive backward locomotion (if technically feasible). The second question could be addressed by varying the temporal properties of MDN optogenetic stimulation, and measuring the effects on behavior or downstream interneuron activity. Either or both experiments would provide the reader with context for interpreting the functional imaging and connectivity experiments, which use optogenetic stimuli on the timescale of hundreds of milliseconds to seconds. Not addressing these issues would be a missed opportunity to understand how and where the dynamics of peristaltic backward locomotion arise. This question is directly relevant for understanding to what degree the MDNs are "command-like", and how descending "commands" are transformed into behaviors that evolve over longer temporal and spatial scales.

---

## [Author Response]

Essential revisions:1a) The paper does not consider the dynamics of backward locomotion, and the role of the MDNs and downstream circuits in generating these dynamics. Reviewer #2 describes 2 experiments that could address this issue. The second experiment (varying the temporal properties of MDN optogenetic stimulation, and measuring the effects on behavior or downstream interneuron activity) seems doable and would definitely enhance the paper but it is not critical, and if it cannot be performed within the timeframe of revisions, this concern should be dealt with as a caveat in the Discussion.

We agree that these are interesting and informative experiments. Please see response to reviewer #2, below.

1b) The statement that "…MDNs coordinate a transition between antagonistic larval locomotor behaviors…" seems an overreach unless some data is presented on the relationship between MDN activity and backward locomotion.

We show that MDNs induce backward locomotion in larvae (Figure 1). We also show that MDNs activate Pair1 neurons (Figure 6), and that Pair1 halts forward locomotion (Figure 7) – and forward and backward locomotion are antagonistic behaviors. Finally, new data requested by reviewer #2 shows that MDN activity triggers backward locomotion (see new Figure 2F). So it seems justified to say that MDNs coordinate a transition between antagonistic larval behaviors.

2a) Command is always a sticky issue because it is often loosely used. The authors provide a nice sufficiency test but perform a necessity test only for backward crawling in response to mild mechanical stimulation of the anterior body and suppression is not 100% complete. We suggest that the authors clearly define command, if they wish to maintain this terminology and that they carefully circumscribe its usage.

We now avoid the term “command neuron” altogether, and we define “command-like neuron” in the Abstract as a neuron that can elicit a specific behavior, without a requirement to be fully necessary for the behavior.

2b) Please clarify why the MDNS are not always active during backward crawls as assessed by CaMPARI.

The CaMPARI experiment sums neuronal activity over a relatively long window (30 seconds) and this leads to variability due to different larvae performing different numbers of backward motor waves in the interval. In addition, the MDN axons have lower signal to noise than experiments imaging large dendritic arbors, making low values more likely. Thus, we don’t want to overinterpret any single data point in this experiment, but rather view the population as a whole. Interpreting the data conservatively, as a population, allows us to make the conclusion that MDNs are preferentially active during backwards locomotion. A rigorous determination of MDN activity during a backward wave would require dual color calcium imaging in MDNs and motor neurons (see response to reviewer #2, comment 1, below) rather than a CaMPARI experiment.

Reviewer #2:1) The paper does not consider the dynamics of backward locomotion, and the role of the MDNs and downstream circuits in generating these dynamics. For example, is MDN activity sustained throughout backward locomotion? Is transient activation of the MDNs sufficient to produce sustained backward movement? At the circuit level, the first question could be addressed by imaging from MDN neurons during fictive backward locomotion (if technically feasible).

We agree, but as the reviewer surmised, it is not technically feasible with existing reagents. We tried expressing GCaMP in both MDNs and motor neurons and imaging neuron-specific ROIs, but it was not possible due to the massive overlap in motor dendrites and MDN axon in thorax and abdominal segments; imaging MDN in the brain and motor neurons in the VNC was not possible due to their difference in z-axis position. We also tried imaging MDN and various pre-motor neurons (as a proxy for motor activity) but found the overlapping expression also precluded acquiring neuron-specific data. The ideal experiment would be to express RCaMP in motor neurons using the RN2-LexA driver and GCaMP in the MDNs using the Split3 Gal4 driver. To build this stock would take about two months, plus the time to do the experiment, which is beyond the time allotted for this revision.

The second question could be addressed by varying the temporal properties of MDN optogenetic stimulation, and measuring the effects on behavior or downstream interneuron activity. Either or both experiments would provide the reader with context for interpreting the functional imaging and connectivity experiments, which use optogenetic stimuli on the timescale of hundreds of milliseconds to seconds. Not addressing these issues would be a missed opportunity to understand how and where the dynamics of peristaltic backward locomotion arise. This question is directly relevant for understanding to what degree the MDNs are "command-like", and how descending "commands" are transformed into behaviors that evolve over longer temporal and spatial scales.

We agree, and have performed transient (300ms) activation of MDN and assayed intact larvae or fictive preparations for the number of backward waves. We found that this brief pulse of MDN activity invariably triggers one wave of backward larval locomotion, in both intact and fictive preparations, with fictive preps sometimes having a second wave at variable times during the 25 second imaging interval following the induced backward wave. Note that backward waves are common in fictive preps (Pulver et al., 2015), and it is likely that these second waves are simply background backward wave activity, as they are not time-locked to the first (induced) wave. Nevertheless, it is clear from our new experiments that transient (300 millisecond) activation of MDN always leads to a multi-second backward locomotor wave. These data are shown in new Figure 2F.